# Regulated repression governs the cell fate promoter controlling yeast meiosis

Janis Tam [1] & Folkert J. van Werven [1]✉

Intrinsic signals and external cues from the environment drive cell fate decisions. In budding yeast, the decision to enter meiosis is controlled by nutrient and mating-type signals that regulate expression of the master transcription factor for meiotic entry, *IME1*. How nutrient signals control *IME1* expression remains poorly understood. Here, we show that *IME1* transcription is regulated by multiple sequence-specific transcription factors (TFs) that mediate association of Tup1-Cyc8 co-repressor to its promoter. We find that at least eight TFs bind the *IME1* promoter when nutrients are ample. Remarkably, association of these TFs is highly regulated by different nutrient cues. Mutant cells lacking three TFs (Sok2/Phd1/Yap6) displayed reduced Tup1-Cyc8 association, increased IME1 expression, and earlier onset of meiosis. Our data demonstrate that the promoter of a master regulator is primed for rapid activation while repression by multiple TFs mediating Tup1-Cyc8 recruitment dictates the fate decision to enter meiosis.

[1] Cell Fate and Gene Regulation Laboratory, The Francis Crick Institute, 1 Midland Road, London NW1 1AT, UK. ✉email: folkert.vanwerven@crick.ac.uk

The choice of whether to differentiate into another cell type is directed by multiple cell intrinsic and extrinsic environmental factors. These cues signal to master regulatory genes, which in turn control the initiation of cell differentiation programmes. As a result, multiple signals are transformed into a binary decision: whether to undergo cell differentiation or not.

Budding yeast cells undergo a differentiation programme called gametogenesis or sporulation during which a diploid cell gives rise to four haploid spores. Yeast gametogenesis is characterised by one round of DNA replication and recombination, two consecutive chromatin segregation events called meiosis followed by spore formation[1]. As a result, an ascus with four haploid spores is produced. In yeast, the decision to enter meiosis is controlled by a master regulatory transcription factor (TF) named inducer of meiosis 1, IME1[2,3]. In the absence of IME1, cells cannot enter meiosis and produce gametes. Thus, understanding how IME1 is regulated is key to understanding how the decision to enter meiosis is made.

Multiple transcriptional control mechanisms regulate IME1 expression. The IME1 gene has an unusually large promoter for the yeast genome (over 2.4 kb) that integrates multiple signals[4]. Nutrient and mating type signals ensure that IME1 is only expressed in the appropriate nutrient environment and in the correct cell type. Only cells harbouring opposite mating-type loci (MATa and MATα) can induce IME1[5,6]. In cells with a single mating type (MATa or MATα), the TF Rme1 is expressed and induces transcription of the long noncoding RNA (lncRNA) IRT1, which in turn transcribes through the IME1 promoter and represses IME1 expression[7]. In MATa/α diploid cells, a second lncRNA upstream of IRT1 named IRT2 interferes with IRT1 transcription, thus forming a positive feedback loop by which Ime1 promotes its own expression[8].

In order to induce IME1 transcription, diploid cells must be starved for glucose and nitrogen, and cells need to be respiring[4,9]. The glucose and nitrogen signals integrate at the IME1 promoter. Distinct sequence element mediates IME1 repression by glucose signalling, while other parts of the promoter respond to nitrogen availability[10]. Notably, the TF Sok2 controls IME1 promoter activity via the glucose responding element[11]. Multiple other TFs contribute to regulation of IME1 transcription[12–14]. Moreover, over 50 TFs have a conserved consensus site in the IME1 promoter and about 30 TFs may directly or indirectly control IME1 transcription[12].

The nutrient control of IME1 expression is mediated by multiple signalling pathways, including PKA, TOR complex 1 (TORC1), AMP-activated protein kinase (AMPK) and mitogen-activated protein kinase (MAPK)[15–17]. Inhibiting two signalling pathways, PKA and TORC1, is sufficient to induce IME1 expression in cells exposed to a nutrient rich environment where IME1 expression is normally repressed[16]. Thus, PKA and TORC1 signalling is essential for controlling IME1 expression and hence the decision to enter meiosis (Fig. 1a). Previously, we showed that Tup1 represses the IME1 promoter under nutrient rich conditions[16]. Tup1 is part of the Tup1–Cyc8 co-repressor complex, which is involved in repression of more than 300 gene promoters in yeast[18–20]. During starvation, when PKA and TORC1 activity is reduced, Tup1 dissociates from the IME1 promoter and IME1 transcription is concomitantly induced.[16]. How Tup1–Cyc8 association with the IME1 promoter is regulated may be key to how IME1 promoter activity is controlled.

Here, we report how the Tup1–Cyc8 co-repressor complex regulates IME1 transcription. In short, we found that regulated repression by multiple sequence specific TFs mediating the association of Tup1–Cyc8 with the IME1 promoter is the means by which IME1 transcription is controlled. Our data indicate that nutrient cues regulate the association of Tup1–Cyc8 interacting TFs with the IME1 promoter, which is key to regulating IME1 expression. Our work provides a framework for understanding how nutrient signals integrate at a cell fate promoter and control a critical developmental decision in yeast.

## Results

**Tup1–Cyc8 prevents activation of the IME1 promoter.** Previously, we reported that Tup1 associates between 800 and 1400 base pairs (bp) upstream of the IME1 translation start site[16]. If the region of the IME1 promoter where Tup1 binds is also important for IME1 activation, then deleting that part of the promoter should affect the onset of meiosis. We generated six truncation mutants with a 200 bp interval in the IME1 promoter and examined the ability of these mutants to undergo meiosis (Fig. 1b). The largest truncation mutant that underwent meiosis with comparable kinetics as wild-type cells harboured 1400 bp of the IME1 promoter (pIME1(−1400-2315Δ)) indicating that this region harbours the regulatory elements required for complete activation of the IME1 promoter (Fig. 1b). In addition, we found that meiosis in pIME1(−800-2315Δ) was completely impaired, whereas pIME1(−1200-2315Δ) had a much milder effect on meiosis. The result suggests that a region between −800 and −1200 bp harbours regulatory elements essential for IME1 promoter function. Analysis of additional mutants revealed that the region between −800 and −850 bp contains regulatory elements important for IME1 activation (Supplementary Fig. 1). In conclusion, the region essential for Tup1 binding to the IME1 promoter is also required for transcription of IME1.

Tup1 forms a complex with Cyc8[19,20]. The Tup1–Cyc8 co-repressor complex is conserved and plays various roles in regulating gene transcription[21]. Like Tup1, Cyc8 has also been implicated in regulation of IME1 expression[22]. To investigate how Cyc8 regulates IME1 expression, we determined Cyc8 binding with the IME1 promoter under nutrient rich conditions. We found that Cyc8 peaked in the same region as Tup1 in the IME1 promoter (Figs. 1c and 1d). These data indicate that Tup1–Cyc8 regulates the IME1 promoter.

Various models for Tup1–Cyc8 mediated repression of target gene promoters have been described[21,23,24]. It has been proposed that Tup1–Cyc8 primarily regulates promoters by masking TFs from recruiting co-activators[25]. If Tup1–Cyc8 represses the IME1 promoter by shielding co-activators, then transcriptional activators should be present at the promoter under repressive conditions. To test this, we measured the association of a known transcriptional activator of IME1, Pog1[7]. We found that Pog1 is indeed enriched at the IME1 promoter under repressive conditions (Fig. 1d). To further examine whether transcriptional activators are readily present at the IME1 promoter, we measured IME1 expression after depletion of Tup1 or Cyc8. We reasoned that if Tup1–Cyc8 represses the IME1 promoter by restraining activating TFs, then depletion of Tup1 or Cyc8 should concomitantly allow activators present to induce IME1 transcription. We used the auxin inducible degron (AID) system (TUP1-AID and CYC8-AID), and treated cells with indole-3-acetic acid (IAA) to achieve rapid protein depletion in cells[26]. Rapid and sustained depletion of Tup1 and Cyc8 was achieved within 30 min after IAA treatment (Fig. 1e). Strikingly, IME1 transcript levels strongly increased concurrently, and were comparable to or even higher than those in wild-type cells entering meiosis when IME1 expression is typically at its peak (Fig. 1f). These data show that Tup1–Cyc8 represses the IME1 promoter under nutrient rich conditions and suggest that the default state of the IME1 promoter is active.

Although Pog1 is already bound at the IME1 promoter in repressive conditions, it is possible that other transcriptional

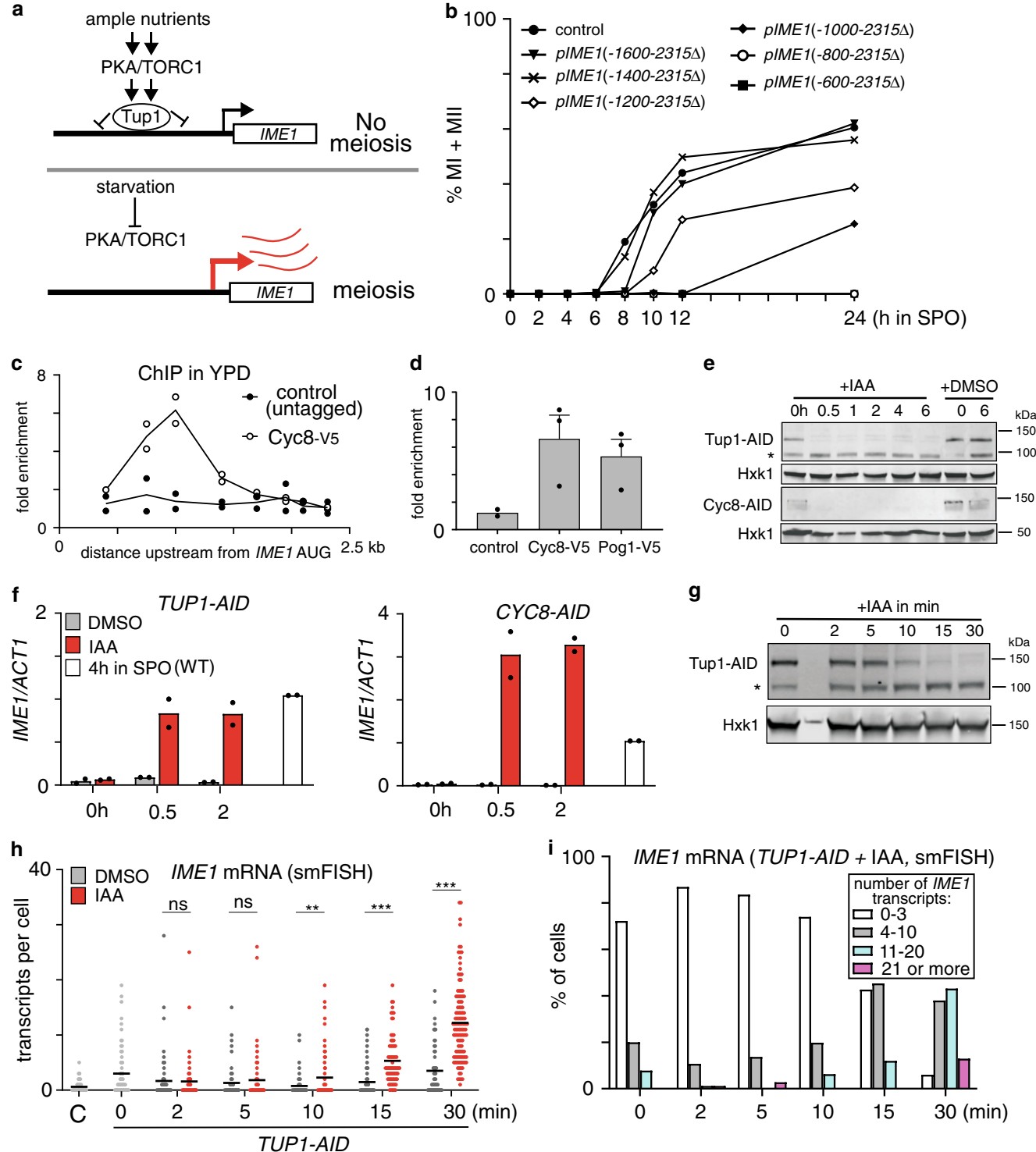

activators associate with the *IME1* promoter after Tup1–Cyc8 dissociates. This may result in a delay between Tup1–Cyc8 depletion and activation of *IME1* transcription. We therefore decided to monitor *IME1* expression by single molecule RNA fluorescence in situ hybridisation (smFISH)[27]. We found that as soon as Tup1 was depleted, *IME1* transcripts were detected in single cells (Fig. 1g–i, Supplementary Fig. 2a). An increase in *IME1* transcripts was detected as early as 10 min after IAA treatment when Tup1 was partially depleted (Fig. 1g). After 15 min, 5.3 *IME1* transcripts were detected per cell on average, and

12% of the cells (*TUP1-AID* + IAA) had more than 10 *IME1* transcripts compared to 2% in control cells (Fig. 1i). After 30 min, more than 55% of cells expressed more than 10 *IME1* transcripts (Fig. 1i). The AID-tag fused to Tup1 had some effect on *IME1* expression in the absence of IAA as *IME1* transcript levels were increased by five-fold in *TUP1-AID* compared to wild-type cells (Fig. 1h). Our analysis indicates that there is little temporal delay between Tup1–Cyc8 depletion and *IME1* transcript accumulation suggesting that transcriptional activators are readily available for activating the *IME1* promoter.

**Fig. 1 Tup1–Cyc8 prevents activation of the *IME1* promoter. a** Schematic representation of nutrient control of the *IME1* promoter. **b** Effects of truncations in the *IME1* promoter on meiosis. Diploid cells with one copy of *IME1* deleted (control, FW4128) and harbouring promoter truncations at the WT *IME1* copy (*pIME1(−1600-2315Δ)*, FW3946; *pIME1(−1400-2315Δ)*, FW3947; *pIME1(−1200-2315Δ)*, FW3948; *pIME1(−1000-2315Δ)*, FW3949; *pIME1(−800-2315Δ)*, FW3950; *pIME1(−600-2315Δ)*, FW3951) were induced to enter meiosis. Samples were taken at the indicated time points, fixed and DAPI masses were counted ($n = 200$ cells per sample) to determine meiosis (MI + MII). **c** Cyc8 binding to *IME1* promoter determined by chromatin immunoprecipitation (ChIP). Cyc8 bound DNA fragments were isolated and quantified by qPCR using eight different primer pairs from cells expressing V5 epitope-tagged Cyc8 (FW6381). The signals were normalised over *HMR*. Mean of $n = 2$ is shown. **d** Similar as **c** except that the binding of Cyc8 (Cyc8-V5, FW6381) and Pog1 (Pog1-V5, FW968) is shown alongside untagged wild-type (control, FW1511) cells. The region around 1000 bp upstream of the *IME1* AUG was analysed. Mean of $n = 3$ (control is $n = 2$) and SEM are shown. **e** Tup1 and Cyc8 depletion detected by western blot. Diploid cells harbouring Tup1 or Cyc8 fused to auxin induced degron (AID) (*TUP1-AID*, FW5057; *CYC8-AID*, FW6371) were treated with IAA or DMSO. As a control, Hxk1 levels were determined. *Tup1-AID cleavage product with no detectable function. A representative AID depletion experiment is shown ($n > 3$). **f** Similar as **e** except that *IME1* mRNA expression was determined by RT-qPCR. Mean of $n = 2$ is shown. **g** Tup1 protein levels (*TUP1-AID*, FW5057) detected by western blotting as described in (**e**). *Tup1-AID cleavage product with no detectable function. A representative AID depletion experiment is shown ($n > 3$). **h** Distribution of *IME1* transcript levels in single cells as described in **g** determined by single molecule RNA fluorescence in situ hybridisation (smFISH). Cells were hybridised with *IME1* (AF594) and *ACT1* (Cy5) probes. Cells positive for *ACT1* were used for the analyses. Data of $n \geq 50$ cells and mean (black line) are displayed. Unpaired parametric two-tailed Welch's *t* test with 95% confidence was used. Non-significant (ns) and *p* values ($** = \leq 0.01$, $*** = \leq 0.001$) are indicated. **i** Same as **h** with data binned by expression levels.

**Multiple Tup1–Cyc8 interacting TFs bind to the *IME1* promoter**. The Tup1–Cyc8 complex interacts with DNA sequence specific TFs to form co-repressor complexes at promoters[28–34]. These TFs facilitate Tup1–Cyc8 association with promoters and mediate repression of target genes. To investigate which TFs recruit Tup1–Cyc8 to the *IME1* promoter and how they control *IME1* transcription, we assembled a list of TFs previously reported to interact with Tup1 or Cyc8. In addition, we examined the region of the *IME1* promoter (−600 to −1200 bp) where Tup1–Cyc8 binds and scanned for sequence motifs among TFs known to interact with Tup1–Cyc8 (Fig. 2a, Supplementary Fig. 3). We identified 13 candidate TFs that were known or implicated to interact with Tup1–Cyc8 and have binding sites in the *IME1* promoter (Fig. 2a, Supplementary Fig. 3). We also included the TF Sok2 in our analyses because it has been proposed to interact with Tup1–Cyc8 and Sok2 is known to directly repress *IME1* transcription[11,28]. After the curation of the list of TFs, we measured their binding under nutrient rich conditions. Eight TFs displayed enrichment (three-fold or more over background) at the *IME1* promoter (Fig. 2b). As expected, a known regulator of the *IME1* promoter, Sok2, was strongly enriched[11]. Phd1 (a paralogue of Sok2) and Yap6 also displayed strong enrichment (Fig. 2b). In addition, the TF Sut1, which is known to interact with Cyc8, was enriched[31]. The TFs Mot3, Sko1, Nrg1 and Nrg2 displayed a milder enrichment (between three-fold and six-fold). For the TFs that displayed enrichment, we also assessed their binding to other parts of the *IME1* promoter (Fig. 2c). The binding of these TFs peaked in the same region of the *IME1* promoter as Tup1–Cyc8. Thus, at least eight TFs that have been implicated to interact with Tup1–Cyc8 associate with the *IME1* promoter.

Next, we examined whether these TFs are responsible for recruiting Tup1 to the *IME1* promoter. We reasoned that candidate TFs should associate independently of Tup1–Cyc8 to the *IME1* promoter, while the binding of Tup1–Cyc8 should depend on the TFs and thus the presence of their binding motifs (Fig. 3a). First, we depleted Tup1 (*TUP1-AID* + IAA) and measured binding for a subset of the bound TFs (Fig. 3b, Supplementary Fig. 4a). Except for Sko1, the binding of the TFs to the *IME1* promoter was not affected by Tup1 depletion (Fig. 3b). Thus, multiple TFs known to interact with Tup1–Cyc8 associate with the *IME1* promoter independently of Tup1–Cyc8. Second, we examined whether the candidate TFs contribute to Tup1–Cyc8 recruitment. We mutated putative binding sites of the TFs that showed binding to the *IME1* promoter. To do so, we generated a construct containing the full-length promoter controlling the

expression of a sfGFP fused to the *IME1* gene (*pIME1-WT*). Subsequently, we mutated 103 nucleotides distributed across a region of 400 bp in the *IME1* promoter where most TF binding sites were present (*pIME1-bsΔ*) (Fig. 3c, Supplementary Fig. 4b, c). We integrated the constructs into the *TRP1* locus in cells harbouring a deletion of the endogenous *IME1* gene and promoter sequence. Tup1 association with the *IME1* promoter was nearly completely lost in *pIME1-bsΔ* cells (Fig. 3c). As expected, control cells (*pIME1-WT*) displayed strong Tup1 binding. Finally, we assessed how *IME1* expression is affected in *pIME1-bsΔ* cells. Surprisingly, *IME1* expression in *pIME1-bsΔ* cells was reduced, suggesting the regulatory elements essential for Tup1–Cyc8 recruitment are also important for *IME1* activation (Fig. 3d). In conclusion, DNA sequence motifs of TFs bound to the *IME1* promoter are required for Tup1–Cyc8 association with the *IME1* promoter.

**Tup1–Cyc8 does not directly control the chromatin state**. The Tup1–Cyc8 complex represses gene promoters, at least in part, by stabilising nucleosomes and establishing repressive chromatin[35–37]. Indeed, the region where Tup1–Cyc8 binds in the *IME1* promoter is devoid of nucleosomes in the absence of Tup1[16,35]. One possibility is that the repressive effects of Tup1–Cyc8 on the *IME1* promoter are indirect. Instead of Tup1–Cyc8 directly promoting nucleosome occupancy or stability, nucleosomes could be evicted by transcriptional activators together with chromatin remodellers in the absence of Tup1–Cyc8. Such an activator–repressor model has been proposed for Tup1–Cyc8 and functionally demonstrated for other loci[25]. Since *pIME1-bsΔ* cells displayed low binding of Tup1 and no de-repression of *IME1* expression (Figs. 3c and 4a), we can discriminate between direct and indirect effects of Tup1–Cyc8 in regulating chromatin state. We found that in *pIME1-bsΔ* cells, relative histone H3 occupancy was slightly increased in the region between 400 and 1000 bp upstream of *IME1* AUG and mostly unaltered in the rest of the *IME1* promoter when compared to *pIME1-WT* cells grown in rich medium (Fig. 4b, Supplementary Fig. 5a). Conversely, under activating conditions, histone H3 was strongly reduced in *pIME1-WT* cells but to a significantly lesser degree in *pIME1-bsΔ* cells (Supplementary Fig. 5a, bottom panel).

Tup1–Cyc8 also directly interacts with class I and II histone deacetylases (HDACs), which in turn confer repression through deacetylation of nucleosomes[38–40]. For example, repression of the *FLO1* promoter is achieved by Tup1–Cyc8 mediated recruitment of Hda1 and Rpd3[41]. In *hda1Δ rpd3Δ* cells strong de-repression

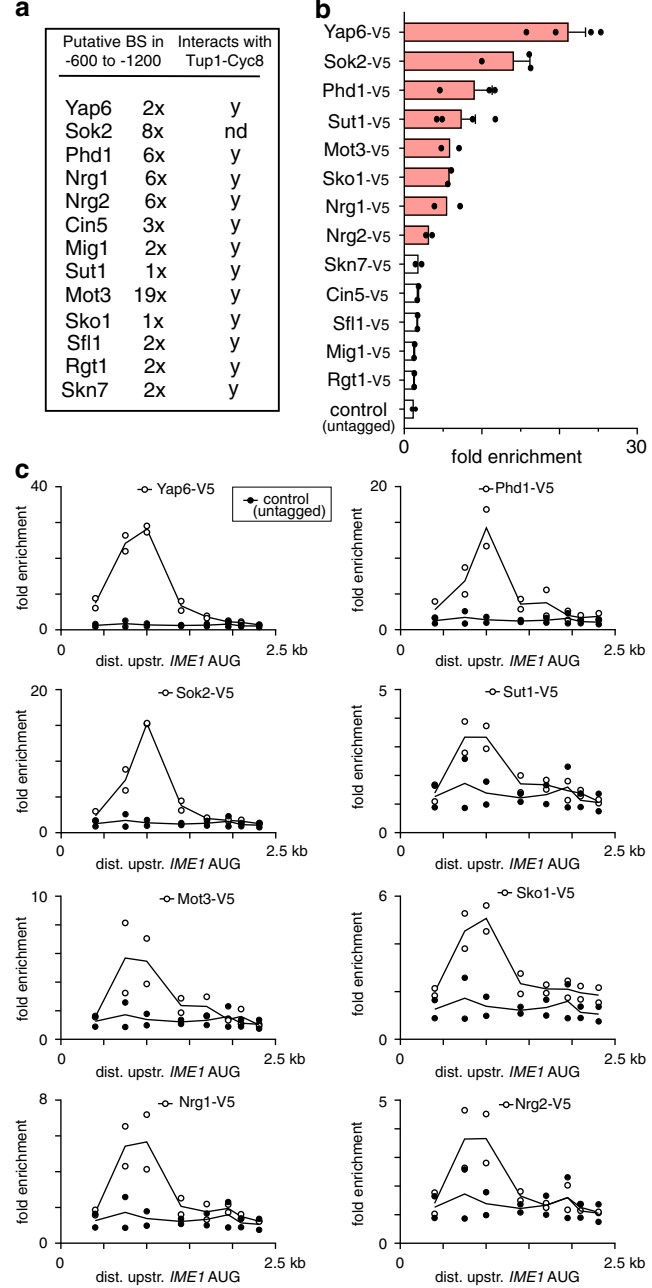

**Fig. 2 Multiple Tup1–Cyc8 interacting TFs associate with the IME1 promoter. a** Table of TFs that have been implicated to interact with Tup1–Cyc8 and have motif sequences within the *IME1* promoter. BS = binding site. **b** Multiple TFs associate with the *IME1* promoter in the same region where Tup1–Cyc8 binds. TFs listed in **a** were tagged with V5 epitope tag. Diploid cells harbouring V5-tagged transcription factor (*YAP6-V5*, FW3833; *SOK2-V5*, FW5638; *PHD1-V5*, FW4466; *SUT1-V5*, FW6974; *MOT3-V5*, FW4383; *SKO1-V5*, FW4389; *NRG1-V5*, FW4393; *NRG2-V5*, FW4396; *SKN7-V5*, FW4399; *CIN5-V5*, FW7072; *SFL1-V5*, FW7070; *MIG1-V5*, FW4665; *RGT1-V5*, FW4386) and control cells (untagged, FW1511) were grown till exponential growth. The region around 1000 bp upstream of *IME1* AUG was analysed for binding by ChIP. The signals were normalised over *HMR*. Red bars represent a signal of three-fold over *HMR* or more. Mean and SEM of n = 4 (Yap6-V5 and Sut1-V5), n = 3 (Sok2-V5 and Phd1-V5) and n = 2 for other TFs plus untagged control are shown. **c** Similar as **b**, except that TFs that showed at least three-fold enrichment in **b** were analysed for binding throughout the whole *IME1* promoter. Eight primer pairs were used for the analyses. The mean of n = 2 is shown.

of the *FLO1* gene can be observed. To examine whether histone deacetylases mediate Tup1–Cyc8 repression at the *IME1* promoter, we generated single and double mutants of known Tup1–Cyc8 interacting histone deacetylases (Rpd3, Hda1, Hos1 and Hos2) and measured *IME1* expression levels by smFISH (Fig. 4c, Supplementary Fig. 5b). Deletion of individual HDACs (*rpd3Δ*, *hda1Δ*, *hos1Δ* and *hos2Δ*) did not increase expression of *IME1*. About 10% of *rpd3Δhda1Δ* cells expressed four or more *IME1* transcripts, which is a marginal increase when compared to Tup1 depleted cells (Supplementary Fig. 5b, Fig. 1i). Two other double mutants (*rpd3Δhos1Δ* and *rpd3Δhos2Δ*) displayed no detectable increase in *IME1* expression but showed a decrease in *IME1* levels similar to that observed in the *rpd3Δ* mutant (Fig. 4c). Our data suggest HDACs that are known to interact with Tup1–Cyc8 play only a marginal role in repressing the *IME1* promoter. It is worth noting that our analysis does not exclude the possibility that there is further redundancy between HDACs in regulating the *IME1* promoter. Taken together, our analyses suggest a marginal role for Tup1–Cyc8 in regulating the *IME1* promoter through chromatin directly.

**Multiple TFs are required for Tup1–Cyc8 mediated repression.** Our analyses of the *IME1* promoter suggest that multiple TFs and binding sites are essential for Tup1–Cyc8 recruitment. Next, we examined how the different TFs control *IME1* expression and mediate Tup1–Cyc8 recruitment. First, we assessed how the paralogues Sok2 and Phd1 control *IME1* expression. *sok2Δ* cells displayed a marginal yet significant increase in *IME1* expression (average transcripts per cell: 0.6 for *sok2Δ* versus 0.3 for control) (Fig. 5a, b). In the *sok2Δphd1Δ* double mutant, *IME1* expression was further increased (average transcripts per cell: 2.2) and about 5% of cells displayed more than 10 transcripts per cell suggesting that Sok2 and Phd1 play redundant roles in tightly repressing the *IME1* promoter (Fig. 5a, b). *IME1* repression was not affected in single or double mutant cells containing *yap6Δ*, but the *sok2Δphd1Δyap6Δ* triple deletion mutant showed the largest increase in *IME1* expression (average transcripts per cell: 2.8 for *sok2Δphd1Δyap6Δ* versus 2.2 for *sok2Δphd1Δ*) (Fig. 5a). About 8% of *sok2Δphd1Δyap6Δ* cells expressed more than 10 *IME1* transcripts per cell (Fig. 5b), which was still much lower than that in cells depleted for Tup1 (Fig. 1i), suggesting that additional TFs contribute to *IME1* repression.

Our data demonstrate that Sok2, Phd1 and Yap6 associate with the *IME1* promoter and contribute to *IME1* repression in nutrient rich conditions. Yet, *IME1* expression was reduced in cells with DNA sequence motifs mutated (*pIME1-bsΔ*). One explanation is that the mutated binding sites in *pIME1-bsΔ* facilitate Tup1–Cyc8 recruitment as well as binding of transcriptional activators. Another possibility is that TFs important for Tup1–Cyc8 recruitment are also required for *IME1* activation. To discriminate between the two possibilities, we generated a construct that contained binding sites for Sok2, Phd1 and Yap6 (*pIME1-spy*), while the other TF binding sites remained mutated (Fig. 5c, Supplementary Fig. 5c). By combining *pIME1-spy* with *sok2Δphd1Δyap6Δ*, we determined whether Sok2, Phd1 and Yap6 are important for *IME1* activation or repression. Tup1 binding was restored in cells harbouring *pIME1-spy* (Fig. 5c). Furthermore, Yap6, Sok2 and Phd1 were enriched at the *IME1* promoter in *pIME1-spy* cells but their binding was reduced compared to the wild-type promoter—suggesting that additional binding sites are present or cooperative interactions with remaining TFs exist (Fig. 5c). Next, we measured *IME1* expression in *sok2Δphd1Δyap6Δ* mutant cells harbouring *pIME1-spy* (Fig. 5d). We found that *IME1* expression was significantly de-repressed—about 17% of cells harbouring *pIME1-spy* and *sok2Δphd1Δyap6Δ* expressed more than 10 *IME1* transcripts per cell (Fig. 5d, e).

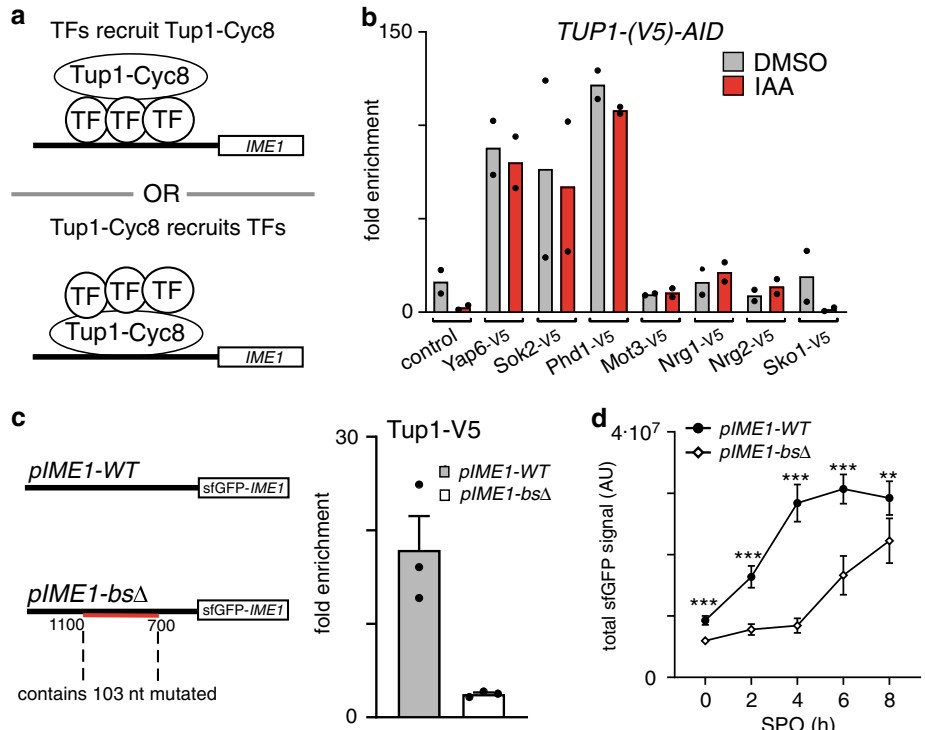

**Fig. 3 Tup1–Cyc8 is recruited by TFs associated with the IME1 promoter. a** Schematic displaying how TFs and Tup1–Cyc8 could interact at the *IME1* promoter. TF = transcription factor. **b** Diploid cells harbouring *TUP1-AID* and V5-tagged TFs (*YAP6-V5*, FW4214; *SOK2-V5*, FW4218; *PHD1-V5*, FW5056; *MOT3-V5*, FW4229; *NRG1-V5*, FW4230; *NRG2-V5*, FW5055; *SKO1-V5*, FW4224) were grown to exponential phase. As a control, *TUP1-AID* (FW5057) cells were also included, which also harbour a V5 tag. Cells were either treated with IAA or DMSO, and the binding of each transcription factor was determined by ChIP and normalised over *HMR*. Mean of $n = 2$ is shown. **c** Binding of Tup1 is affected in cells lacking transcription factor binding motifs in the *IME1* promoter. Diploid cells with a chromosomal deletion of the *IME1* locus, and an integrated plasmid that contained the full *IME1* gene fused with sfGFP at the amino terminus and the wild-type promoter (*pIME1-WT*, FW5370) or the same construct with all the candidate motif sequences mutated (*pIME1-bsΔ*, FW5372) were used for the analysis. These cells also expressed *TUP1-V5*. The binding of Tup1 was determined by ChIP. Mean and SEM of $n = 3$ are shown. **d** Expression of Ime1 during entry into meiosis in *pIME1-WT* and *pIME1-bsΔ* cells. Strains described in **c** were grown till saturation in rich medium (YPD), grown for an additional 16–18 h in pre-sporulation medium (BYTA), and subsequently shifted to sporulation medium (SPO). The levels of Ime1 expression were determined by imaging and quantifying the fluorescent signals generated by sfGFP-Ime1 in single cells ($n = 50$ cells per sample). The mean of signals detected and error bars representing the 95% confidence interval are displayed. Unpaired parametric two-tailed Welch's $t$ test with 95% confidence was used and $p$ values (** = $\leq 0.01$, *** = $\leq 0.001$) are indicated.

As expected, the *sok2Δphd1Δyap6Δ* triple deletion only had a mild effect on *IME1* levels in cells expressing *pIME1-WT* or *pIME1-bsΔ*. We conclude that Sok2, Phd1 and Yap6 are important for repression of the *IME1* promoter and play little role in *IME1* activation.

The Tup1–Cyc8 complex dissociates from the *IME1* promoter in cells exposed to nutrient starvation[16]. We hypothesised that TFs interacting with Tup1–Cyc8 at the *IME1* promoter control Tup1–Cyc8 dissociation during *IME1* activation. To examine this, we measured the binding of the TFs during activation of the *IME1* promoter. In order to induce *IME1* expression and meiotic entry, we typically grow cells in rich medium conditions containing glucose until saturation, then shift to pre-sporulation medium containing acetate to ensure that cells are not subjected to repressive glucose signalling to the *IME1* promoter[4,9]. Subsequently, cells are starved in sporulation (SPO) medium (0.3% acetate), which induces *IME1* transcription and meiotic entry. Both Pog1 and Tup1 were enriched at 0 h in SPO prior to *IME1* induction. As expected, during meiotic entry (4 h in SPO) Tup1 dissociated from the *IME1* promoter completely while Pog1 binding was maintained albeit to a reduced level (Fig. 6a, right panel). In addition, we found that all eight TFs were enriched at the *IME1* promoter prior to induction of *IME1* (Fig. 6a, left panel).

Upon entry into meiosis (4 h in SPO), five TFs showed near background binding (less than three-fold) to the *IME1* promoter, while three TFs (Yap6, Phd1 and Nrg1) displayed marginal enrichment (less than five-fold over background) (Fig. 6a, right panel). We further performed a time course experiment with three TFs (Yap6/Sok2/Phd1) and Tup1. We found that *IME1* bulk expression levels peaked at 3 h in SPO (Supplementary Fig. 6a). Tup1 and Sok2 dissociated from the *IME1* promoter in the early time points (1 and 2 h in SPO), while the majority of Tup1 and Sok2 was evicted at 4 h in SPO (Fig. 6b). Yap6 and Phd1 displayed reduced binding at the *IME1* promoter at 3 h in SPO and was thus slower than Sok2 and Tup1. We propose that the gradual dissociation of multiple TFs evicts Tup1–Cyc8 from the *IME1* promoter.

One possible mechanism by which Tup1–Cyc8 and TFs dissociate from the *IME1* promoter is by re-localisation to the cytoplasm. Indeed, nutrient signalling via PKA and TORC1 can impact the localisation of several TFs[42,43]. We fused mNeonGreen to Sok2, Phd1, Yap6, Tup1 and Cyc8 (Supplementary Fig. 6b). As expected, Sok2, Phd1 and Yap6 were concentrated in the nucleus. Neither protein abundance in the nucleus nor the nuclear-to-cytoplasmic ratios were altered in cells prior to (0 h in SPO) and during entry into meiosis (4 h in SPO) (Fig. 6c, Supplementary

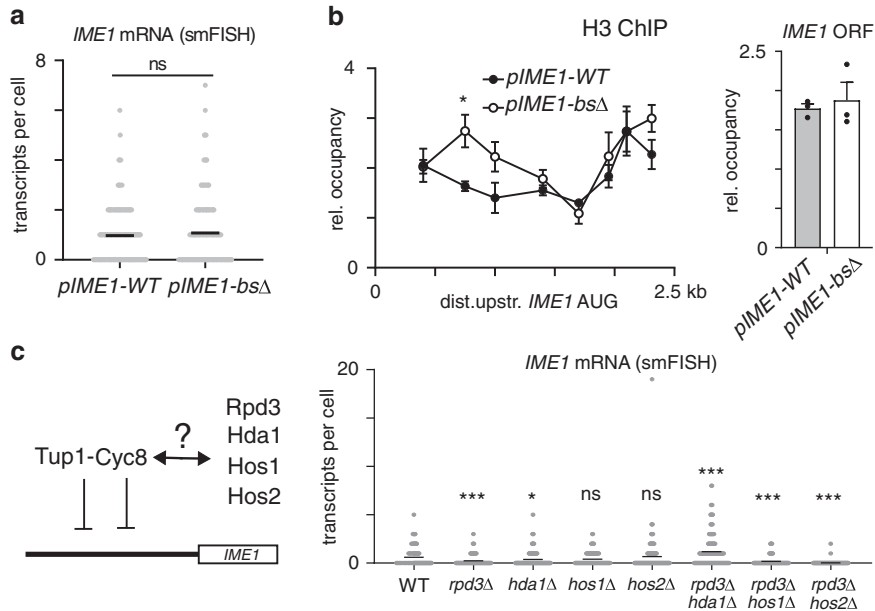

**Fig. 4 Tup1–Cyc8 does not directly regulate chromatin state. a** *IME1* expression in *pIME1-WT* (FW5370) and *pIME1-bs*Δ cells (FW5372) grown to exponential growth in rich nutrient conditions (YPD) detected by smFISH. The mean number of transcripts per cell for each time point (black line) is displayed. At least 200 cells were used for the analysis. Unpaired parametric two-tailed Welch's *t* test with 95% confidence was used. Non-significant (ns) *p* value is indicated. **b** Histone H3 occupancy across the *IME1* promoter and the *IME1* ORF in *pIME1-WT* and *pIME1-bs*Δ cells grown as described in (**a**). The signals were normalised over the *HMR* locus. The mean and SEM of $n = 3$ are shown. * corresponds to a *p* value of ≤0.05 compared to *pIME1-WT* control on a two-way ANOVA followed by a Fisher's LSD test with 95% confidence. **c** Schematic of Tup1–Cyc8 interacting with histone deacetylases (HDACs) (left). *IME1* expression in diploid cells harbouring *rpd3*Δ, *hda1*Δ, *hos1*Δ and *hos2*Δ single deletions (FW8102, FW8426, FW8430 and FW8103) or *rpd3*Δ*hda1*Δ, *rpd3*Δ*hos1*Δ and *rpd3*Δ*hos2*Δ double deletions (FW8457, FW8428 and FW8171) in single cells as determined by smFISH (right). The mean number of transcripts per cell (black line) is displayed. At least 100 cells were used for the analysis. Unpaired parametric two-tailed Welch's *t* test with 95% confidence was used to compare *IME1* expression in mutant cells with wild-type control cells. *p* values (ns = non-significant, * = ≤0.05, *** = ≤0.001) are indicated.

Fig. 6c). Taken together, activation of the *IME1* promoter correlates with dissociation of Tup1–Cyc8 and Tup1–Cyc8 recruiting TFs from the *IME1* promoter, but likely not via a mechanism involving re-localisation of TFs to the cytoplasm.

**Sok2, Phd1 and Yap6 control the onset of meiosis.** Our observations indicate that Sok2, Phd1 and Yap6 are important TFs for *IME1* repression. Next, we investigated how the three TFs control Tup1–Cyc8 recruitment in different nutrient conditions. We found that Tup1 binding to the *IME1* promoter was not decreased by *sok2*Δ, *phd1*Δ and *yap6*Δ single/double/triple deletions in rich medium containing glucose, suggesting that other TFs contribute to *IME1* repression via Tup1–Cyc8 when glucose is used by cells as the carbon source (Fig. 7a). In contrast, prior to meiotic entry (0 h in SPO) Tup1 binding was diminished in *sok2*Δ and *sok2*Δ*phd1*Δ cells, but not in *yap6*Δ and *phd1*Δ cells (Fig. 7a). Strikingly, Tup1 association with the *IME1* promoter was severely reduced (less than three-fold over background) in *sok2*Δ*yap6*Δ and *sok2*Δ*phd1*Δ*yap6*Δ cells at 0 h in SPO. *IME1* expression was inversely correlated with Tup1–Cyc8 recruitment to the *IME1* promoter since *IME1* was significantly de-repressed in *sok2*Δ*yap6*Δ, *sok2*Δ*phd1*Δ or *sok2*Δ*phd1*Δ*yap6*Δ cells at 0 h in SPO (Fig. 7b). Finally, we examined how Sok2, Phd1 and Yap6 mediated Tup1–Cyc8 recruitment contributes to the onset of meiotic entry. Cells harbouring *sok2*Δ or *sok2*Δ*phd1*Δ underwent meiosis much faster than wild-type cells (Fig. 7c). There was little effect on the onset of meiosis in the *yap6*Δ or *yap6*Δ*phd1*Δ mutants. In *sok2*Δ*yap6*Δ and *sok2*Δ*phd1*Δ*yap6*Δ cells the kinetics of meiosis was slightly faster than *sok2*Δ*phd1*Δ cells (Fig. 7c). Approximately 50% of cells underwent meiotic divisions within 2 h in SPO for *sok2*Δ*phd1*Δ*yap6*Δ cells compared to 3 h for

*sok2*Δ*phd1*Δ cells. We also analysed how the onset of meiosis is affected in *sok2*Δ*phd1*Δ*yap6*Δ*nrg1*Δ cells (Supplementary Fig. 7a). We found no difference in the onset of meiosis between *sok2*Δ*phd1*Δ*yap6*Δ*nrg1*Δ and *sok2*Δ*phd1*Δ*yap6*Δ cells, suggesting that Nrg1 is not involved in *IME1* activation (Fig. 6a, right panel). We conclude that Sok2, Phd1 and Yap6 direct Tup1–Cyc8 association with the *IME1* promoter to ensure timely expression of *IME1* in cells grown in acetate-containing medium. Our data further suggest that the *IME1* promoter is regulated by multiple Tup1–Cyc8 co-repressor complexes.

**Binding of TFs is highly regulated by nutrient cues.** Why do so many TFs (at least eight) associate with the *IME1* promoter? One possibility is that the TFs facilitate Tup1–Cyc8 recruitment under different nutrient conditions. With this logic, repression of the *IME1* promoter can be maintained under various nutrient conditions and will only be fully activated when all the nutrient signalling requirements are met. In agreement with this model, *IME1* expression was only marginally increased in *sok2*Δ*phd1*Δ*yap6*Δ cells grown in the presence of ample nutrients with glucose as the carbon source (YPD), and was significantly increased by nearly ten-fold in *sok2*Δ*phd1*Δ*yap6*Δ cells grown in acetate-containing medium (Figs. 5a, b and 7b). Furthermore, in YPD saturation when glucose is depleted from the growth medium, we observed that some TFs displayed altered association with the *IME1* promoter. For example, Sok2 binding was increased, while Yap6 and Cyc8 association with the *IME1* promoter was decreased (Supplementary Fig. 7b). To examine how the different TFs respond to nutrient signalling at the *IME1* promoter more systematically, we measured their association under different nutrient conditions (Fig. 8a). We grew cells until

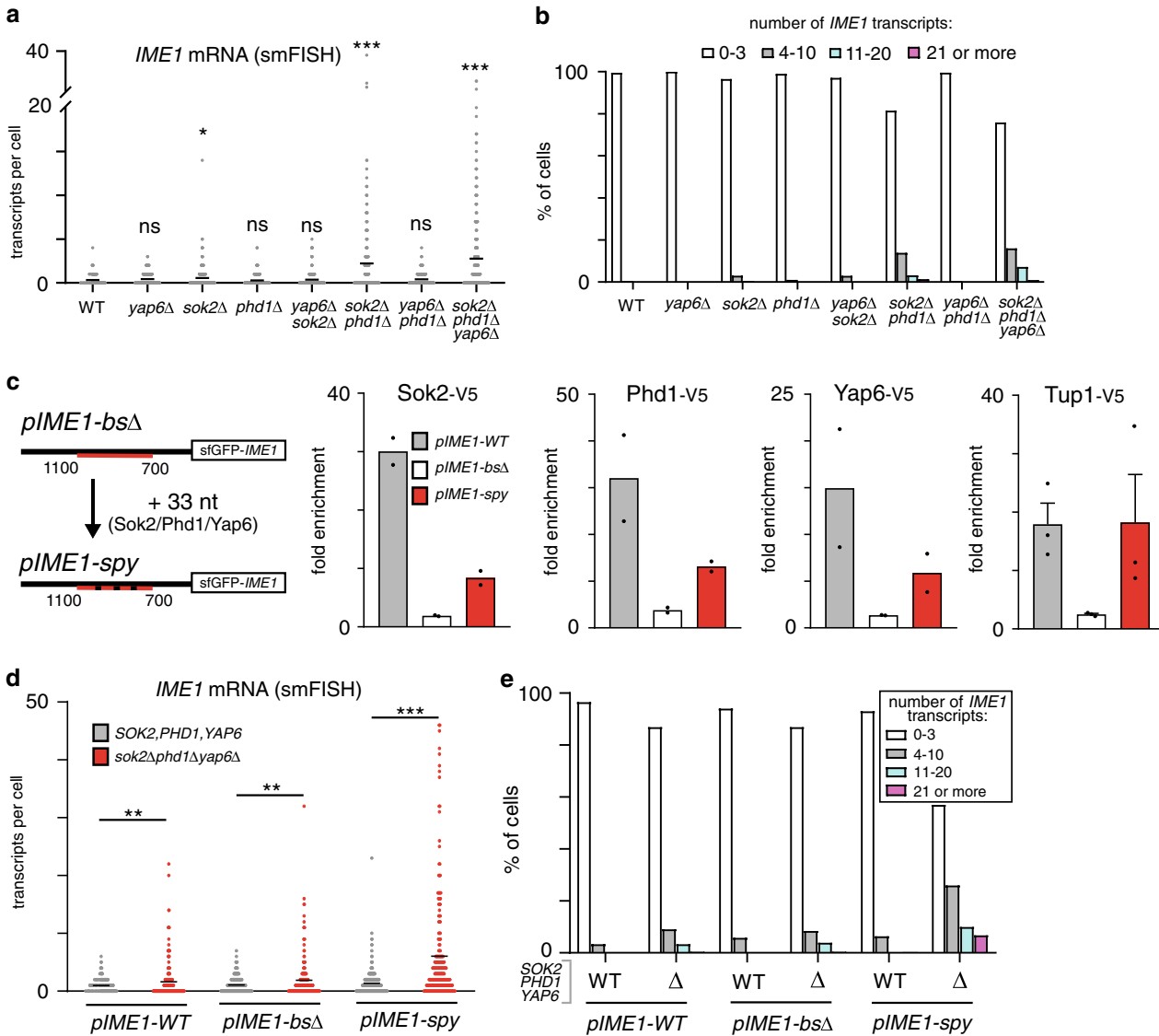

**Fig. 5 Multiple Tup1–Cyc8 interacting TFs repress *IME1* in rich nutrients. a** *IME1* expression in single cells harbouring: *sok2Δ*, *phd1Δ* and *yap6Δ* single deletions (FW3979, FW3991, FW3603); *sok2Δphd1Δ*, *yap6Δphd1Δ* and *sok2Δyap6Δ* double deletions (FW4710, FW4406, FW4239); and *sok2Δphd1Δyap6Δ* triple deletions (FW4010). Cells were grown to exponential phase in nutrient rich conditions (YPD), fixed and hybridised with *IME1* (AF594) and *ACT1* (Cy5) probes. Only cells positive for *ACT1* were used for the analyses. The mean number of transcripts per cell (black line) is displayed. Between 190 and 310 cells were used for the analysis. Unpaired parametric two-tailed Welch's $t$ test with 95% confidence was used to compare *IME1* expression in mutant cells against wild-type control cells. Non-significant (ns) and $p$ values (* = ≤0.05, *** = ≤0.001) are indicated. **b** Same data as in **a**, except that the single cell data for *IME1* were binned according to expression levels. **c** Sok2, Phd1, Yap6 and Tup1 binding in cells harbouring the *pIME1-WT* (FW8081, FW8083, FW8079, FW5370), *pIME1-bsΔ* (FW8087, FW8089, FW8085, FW5372) and *pIME1-spy* (FW8093, FW8095, FW8091, FW7733) constructs as determined by ChIP. Mean of $n = 2$ are displayed, except for Tup1-V5 ($n = 3$) for which the error bars represent SEM values. **d** *IME1* expression in single cells as determined by smFISH in cells harbouring *pIME1-WT*, *pIME1-bsΔ* or *pIME1-spy*. These cells either expressed wild-type *SOK2*, *PHD1* and *YAP6* (FW5370, FW5372, FW7733) or harboured the *sok2Δphd1Δyap6Δ* triple deletions (FW7650, FW8420, FW8177). Unpaired parametric two-tailed Welch's $t$ test with 95% confidence was used. $p$ values (** = ≤0.01, *** = ≤0.001) are indicated. **e** Same data as in **d**, except that the single cell data were binned according to *IME1* expression levels.

the pre-sporulation stage, and subsequently shifted cells to sporulation medium (SPO) (1), SPO plus 2% glucose (2), YP (3) or YP plus 2% glucose (YPD) (4). First, we measured Tup1 association with the *IME1* promoter. We found that in SPO plus glucose, Tup1 binding to the *IME1* promoter was partially restored (Fig. 8b). The association of Tup1 with the *IME1* promoter was further increased in YP and was the highest in YPD growth medium. Pog1, the transcriptional activator of *IME1*, was enriched in all four nutrient conditions, but at higher levels in YP and YPD (Fig. 8b). Interestingly, TFs important for Tup1–Cyc8

recruitment to the *IME1* promoter responded to nutrient signals in distinct ways (Fig. 8c). For example, Yap6, Sok2, Sko1 and Nrg1 associated with the *IME1* promoter in response to the nutrient cues present in YP, but not to glucose signals alone. Phd1 binding partially recovered in the presence of glucose and showed the strongest enrichment in cells exposed to YP and YPD. Conversely, glucose signalling, but not YP, maintained association of Mot3 and Nrg2 with the *IME1* promoter (Fig. 8c). Finally, Sut1 association with the *IME1* promoter was restored in YPD only.

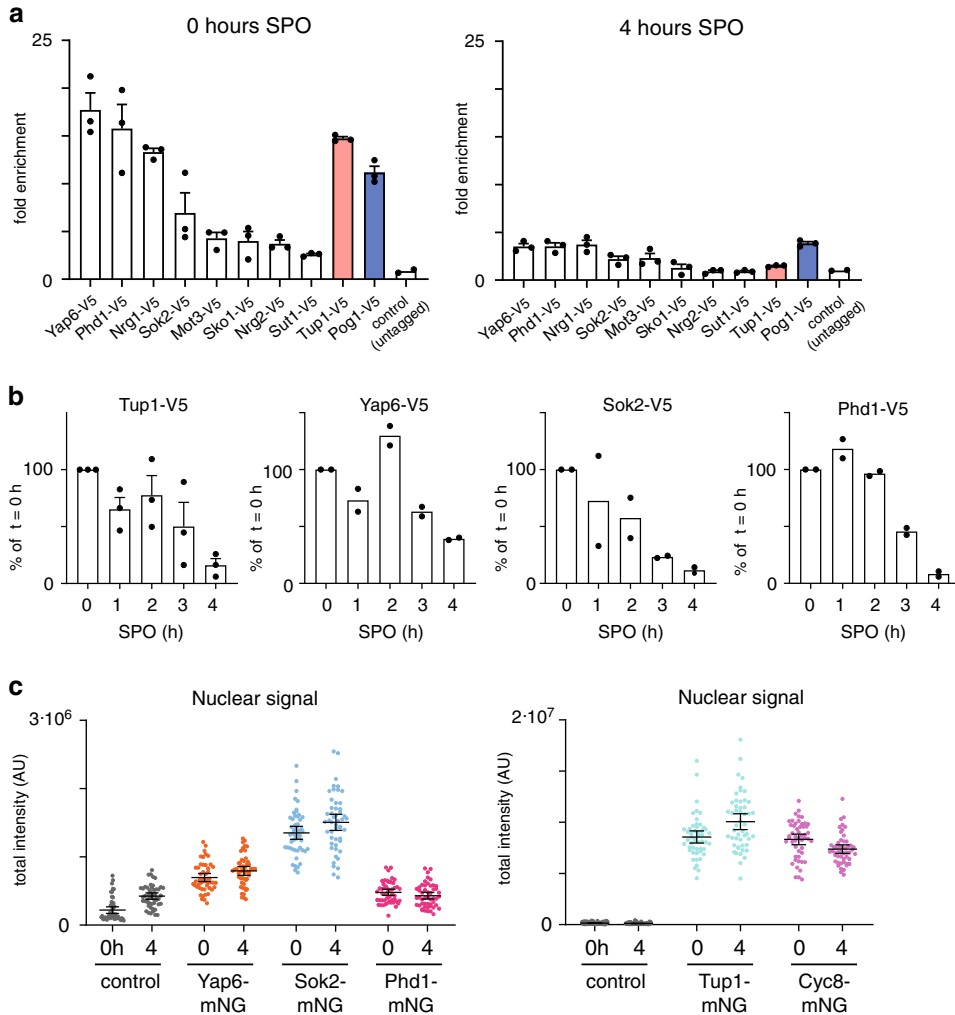

**Fig. 6 Tup1–Cyc8 and TFs dissociate without translocating to the cytoplasm. a** Binding of TFs at the *IME1* promoter prior to and during entry into meiosis. Diploid cells harbouring V5 tagged TFs (*YAP6-V5*, FW3833; *PHD1-V5*, FW4466; *NRG1-V5*, FW4393; *SOK2-V5*, FW5638; *MOT3-V5*, FW4383; *SKO1-V5*, FW4389; *NRG2-V5*, FW4396; *SUT1-V5*, FW6974; *TUP1-V5*, FW3456; *POG1-V5*, FW968) and control cells (untagged, FW1511) were used for the analyses. Samples for ChIP were taken at 0 and 4 h in SPO. Mean and SEM values of $n = 3$ is displayed, except for control ($n = 2$). **b** Similar as in **a**, except that multiple time points were taken to analyse the timing of dissociation of Tup1-V5, Yap6-V5, Sok2-V5 and Phd1-V5 from the *IME1* promoter. ChIP signals were normalised over *HMR*, and the 0 h time point was set to 100%. The mean and SEM values of $n = 3$ (Tup1–V5), and the mean of $n = 2$ (Yap6-V5, Sok2-V5 and Phd1-V5) are displayed. **c** Nuclear signal of Yap6-mNG (FW7473), Sok2-mNG (FW7475), Phd1-mNG (FW7477), Tup1-mNG (FW7644) and Cyc8-mNG (FW7642) prior to (0 h in SPO) and during entry into meiosis (4 h in SPO). Each transcription factor was fused to mNeonGreen (mNG). These cells also expressed mCherry fused to SV40 nuclear localisation signal (NLS) (mCherry-NLS). As a control, the signals of cells harbouring no mNG-tag (FW5199) are displayed. The black bar indicates the mean signal, and each point indicates a single cell measurement ($n = 50$ for each sample). The error bars represent the 95% confidence interval.

Given that Sok2, Phd1 and Yap6 were strongly enriched in cells exposed to YP medium (Fig. 8c), we hypothesised that Tup1–Cyc8 association with the *IME1* promoter is affected in *sok2Δphd1Δyap6Δ* cells in YP, but not in SPO containing glucose. We therefore examined how Tup1–Cyc8 association with the *IME1* promoter was affected in *sok2Δphd1Δyap6Δ* cells under different nutrient conditions. Indeed, in *sok2Δphd1Δyap6Δ* cells, Tup1 binding was detected in SPO plus glucose, but not in YP medium (Fig. 8d). These data indicate that Sok2, Phd1 and Yap6 are important for mediating Tup1–Cyc8 association in YP, while other TFs are required for glucose signalling to the *IME1* promoter (Fig. 8e). In summary, our analyses revealed that the association of one set of TFs (i.e., Mot3 and Nrg2) with the *IME1* promoter is induced by glucose signalling, while another set of TFs (i.e., Yap6, Sok2, Phd1, Sko1 and Nrg1) was recruited to the *IME1* promoter primarily in response to the nutrient cues in YP.

Thus, only when all the required nutrient signalling pathways are repressed, all TFs interacting with Tup1–Cyc8 dissociate from the *IME1* promoter allowing activation of *IME1* transcription. These data show that TFs important for Tup1–Cyc8 recruitment to the *IME1* promoter respond to different environmental cues to ensure Tup1–Cyc8 mediated repression under various nutrient conditions.

## Discussion

We report that the Tup1–Cyc8 complex together with multiple sequence-specific TFs constitute the essential components that control repression of the *IME1* promoter. The decision to enter meiosis and produce gametes is remarkably simple in yeast: environmental signals regulate the association and dissociation of TFs that recruit Tup1–Cyc8 to the *IME1* promoter. We propose that regulated repression of *IME1* by multiple TFs and

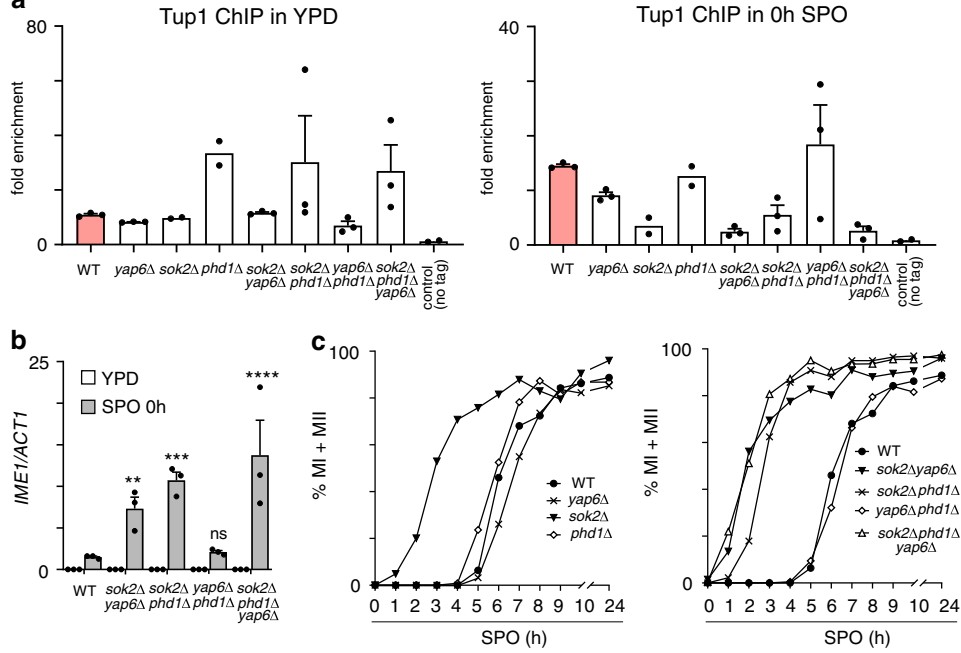

**Fig. 7 Yap6, Sok2 and Phd1 direct *IME1* repression to control meiotic entry. a** Tup1 binding at the *IME1* promoter in nutrient rich conditions (YPD) and prior to entry into meiosis (0 h in SPO) in wild-type cells (FW3456) or cells harbouring *sok2Δ*, *phd1Δ* and *yap6Δ* single deletions (FW3979, FW3991, FW3603); *sok2Δphd1Δ*, *yap6Δphd1Δ* and *sok2Δyap6Δ* double deletions (FW4710, FW4406, FW4239); and *sok2Δphd1Δyap6Δ* triple deletions (FW4010). An untagged control (FW1511) was also included in the analyses. The mean and SEM values of $n = 3$ is displayed, except for *sok2Δ*, *phd1Δ* and the untagged control for which the mean of $n = 2$ is displayed. **b** *IME1* expression in deletion mutants described in (**a**). The signals were normalised over the *ACT1* gene signals. Mean and SEM values of $n = 3$ are displayed. Two-way ANOVA analysis was carried out using the uncorrected Fisher's LSD method with 95% confidence to compare *IME1* expression in mutant cells against wild-type control cells at SPO 0 h. Non-significant (ns) and *p* values (** = ≤0.01, *** = ≤0.001, **** = ≤0.0001) are indicated. **c** Meiosis in mutant strains described in (**a**). Samples were taken at the indicated time points, fixed and DAPI masses were counted ($n = 200–300$ cells per sample) to determine the percentage of cells that underwent meiosis (MI + MII). Cells harbouring two, three or four DAPI masses were classified as meiosis.

Tup1–Cyc8 ensures tight control of the decision to enter meiosis in yeast.

Our data show that repression of *IME1* transcription, and not activation, is highly regulated. Depletion of either Tup1 or Cyc8 completely de-repressed *IME1* expression (Fig. 1). We detected little delay between depletion of Tup1 and de-repression of *IME1* expression in the presence of ample nutrients (Fig. 1g–i). From these two observations, we can infer two important features of the *IME1* promoter. First, the transcriptional activators are bound to the *IME1* promoter or readily available prior to activation of *IME1* transcription. Second, key transcriptional activators of the *IME1* promoter can be active under nutrient rich conditions when Tup1–Cyc8 is unbound. We found that the activator Pog1 is bound to the *IME1* promoter prior to activation. A *pog1* mutant only has a mild effect on *IME1* expression indicating that there must be other transcriptional activators controlling *IME1* transcription[7]. Several other transcriptional activators have been implicated in regulating *IME1* transcription[12].

How does Tup1–Cyc8 control the *IME1* promoter? The Tup1–Cyc8 complex regulates transcription of a subset of promoters in yeast[18,44]. Several mechanisms have been described for Tup1–Cyc8 mediated gene repression[21,23,24]. Our data are largely consistent with a model in which Tup1–Cyc8 masks or shields activating TFs from recruiting co-activators at promoters[25,32]. We showed that multiple (at least eight) TFs that are known to interact with Tup1–Cyc8 associate with the *IME1* promoter (Fig. 2b). Our data suggest that these TFs are important for facilitating Tup1–Cyc8 binding but play little role in *IME1* transcriptional activation. First, almost all TFs involved in

Tup1–Cyc8 recruitment dissociated from the *IME1* promoter upon activation of *IME1* transcription (Fig. 6a, right panel). Second, deleting multiple TFs led to activation, but not repression of *IME1* transcription (Fig. 7c, Supplementary Fig. 7a). However, we cannot exclude that the Tup1–Cyc8 recruiting TFs can function as transcriptional activators in some conditions. Indeed, Yap6 and Sok2 have both been implicated as activators of transcription at some promoters[45,46]. In the context of the *IME1* promoter, each TF likely has a designated function in either repression or activation of *IME1* transcription. We propose that multiple TFs are required to recruit the Tup1–Cyc8 co-repressor to the *IME1* promoter. The Tup1–Cyc8 co-repressor complexes, in turn, mask transcriptional activators (which are different from the Tup1–Cyc8 recruiting TFs) and prevent co-activator recruitment.

Under most environmental conditions, the *IME1* promoter must be repressed to prevent cells from inappropriately entering meiosis and forming gametes only unless when cells are starved. We propose that multiple TFs ensure *IME1* repression under various environmental conditions. First, we found that distinct sets of TFs associate with the *IME1* promoter in different nutrient environments (Fig. 8). Second, deleting three TFs (Sok2, Phd1 and Yap6) led to very mild *IME1* expression in rich medium containing glucose (Fig. 5a, b), but *IME1* was almost fully expressed in cells grown in an acetate-containing medium (Fig. 7b). Thus, additional TFs must facilitate Tup1–Cyc8 association with the *IME1* promoter in rich medium containing glucose.

We previously showed that inhibiting PKA and TORC1 is sufficient to drive entry into meiosis[16]. Understanding how PKA

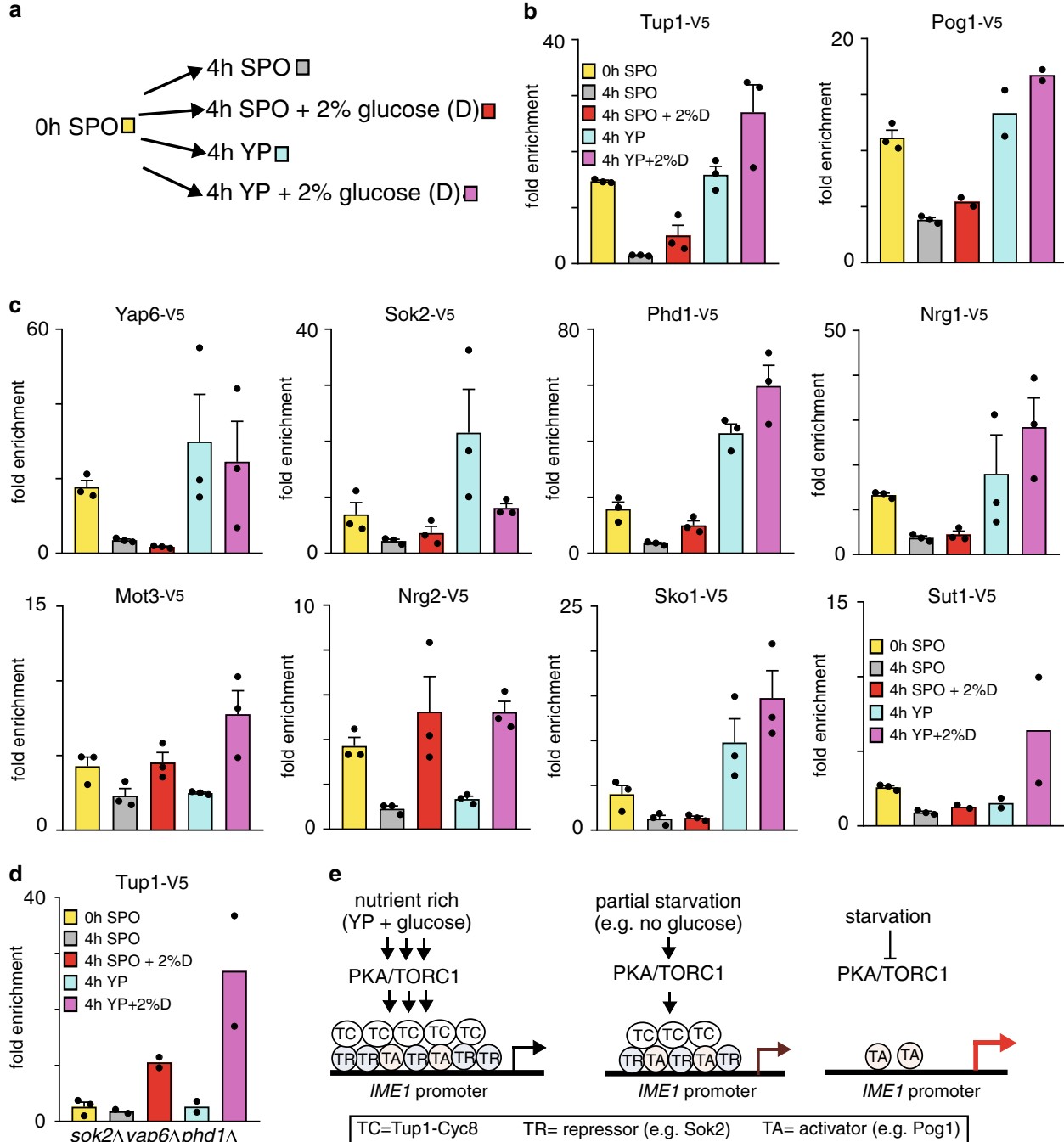

**Fig. 8 Distinct nutrient cues trigger TFs to interact with the *IME1* promoter. a** Schematic of the experimental set up. Following growth in YPD and pre-sporulation medium, cells were shifted to SPO, SPO plus 2% glucose, YP or YP plus 2% glucose for 4 h. **b** Binding of Tup1 and Pog1 at the *IME1* promoter under distinct nutrient conditions described in (**a**), as measured by ChIP. Diploid cells harbouring V5 tagged Tup1 (FW3456) or Pog1 (FW968) were used for the analyses. Normalised mean signals and SEM of $n = 3$ are displayed, except for Pog1-V5 in SPO plus 2% glucose, YP and YPD conditions for which the mean of $n = 2$ is shown. **c** Similar to **b**, except that V5 tagged Yap6, Sok2, Phd1, Nrg1, Mot3, Nrg2, Sko1 and Sut1 were used for the analyses (FW3833, FW5638, FW4466, FW4393, FW4383, FW4396, FW4389 and FW6974). Mean signals and SEM of $n = 3$ are displayed, except for Sut1-V5 for SPO plus 2% glucose, YP and YPD conditions for which the mean of $n = 2$ is shown. **d** Similar to **b**, except that Tup1 binding to the *IME1* promoter was determined in the *sok2Δphd1Δyap6Δ* cells (FW4010). Mean signals for $n = 2$ are displayed, except for 0 h in SPO for which the mean and SEM of $n = 3$ are shown. **e** Model for how nutrient signalling controls the *IME1* promoter. Multiple Tup1–Cyc8 recruiting TFs associate in nutrient rich conditions with the *IME1* promoter. During partial starvation some TFs dissociate from the *IME1* promoter, but Tup1–Cyc8 remains bound. Upon entry into meiosis all TFs and Tup1–Cyc8 dissociate, and consequently *IME1* transcription is induced.

and TORC1 regulate TFs controlling Tup1–Cyc8 recruitment to the *IME1* promoter may be key to understanding how the decision to enter meiosis is regulated. However, the mechanism remains to be deciphered. Inhibiting PKA and TORC1 activity can trigger the re-localisation of TFs, alter protein–protein interactions and protein turnover, and more[32,47]. With this view, we observed no depletion of TFs from the nucleus during activation of the *IME1* promoter (Fig. 6c, Supplementary Fig. 6c). Nutrient signalling may also regulate Tup1–Cyc8 itself through post-translational modifications[48,49].

Our work in yeast shows similarities to how multicellular organisms undertake developmental decisions. In *Drosophila*, plants, and mammals, transcriptional repressors of the Groucho family (structurally related to Tup1) are important for regulating various developmental processes such as body patterning and determination of organ identity[50–52]. Like Tup1–Cyc8, the association of Groucho repressor with promoters relies on sequence specific TFs, and Groucho repressor integrates multiple signals to control gene expression and cell fate outcomes. Regulation of the *IME1* promoter also demonstrates features of enhancer-directed transcriptional control of cell-fate master regulators in mammalian cells[53–55]. Like at the *IME1* promoter, an array of TFs associates and controls the activity of enhancers. In addition, developmentally controlled enhancers are typically regulated by multiple upstream signalling pathways and are often primed for activation[56]. Our findings in yeast may provide insights to better understand how signal integration controls master regulatory genes and developmental decisions in all eukaryotic cells.

## Methods

**Yeast strains and plasmids**. The *Saccharomyces cerevisiae* SK1 genetic background was employed for all experiments in this study. Experiments were carried out with diploid cells and the list of yeast strains described in this study can be found in Supplementary Data 1. Gene deletions, *IME1* promoter truncations, and protein fusions were achieved by the single step PCR-based gene modification protocol described in ref. [57]. Auxin-based depletion of Tup1 and Cyc8 was achieved by fusing Tup1 and Cyc8 with the auxin-induced degron (AID) tag containing 3×V5 epitope and the *Arabidopsis thaliana* IAA7 protein[26]. The *Oryza sativa* TIR1 ligase (*osTIR1*) was also expressed under the *TEF1* promoter from a plasmid integrated at the *HIS3* locus (courtesy of Leon Chan) in the *TUP1-AID* and *CYC8-AID* strains. Protein-mNeonGreen fusions were constructed by tagging the proteins with mNeonGreen tagging cassettes (courtesy of Andreas Doncic) described in ref. [58]. Cells expressing protein-mNeonGreen fusions also harboured a nuclear localisation signal peptide derived from simian virus 40 tagged with two copies of mCherry (2xmCherry-SV40NLS)[58].

Single-copy integration plasmids containing *IME1* N-terminally fused with *sfGFP* and full length *IME1* promoter were derived from *pNH604*[59]. The *pIME1-sfGFP-IME1* fragment (~4.6 kb) was amplified from yeast cells expressing sfGFP-Ime1 and was cloned into *pNH604* plasmid in the NotI and BamHI sites by restriction digestion. The resulting plasmid (*pIME1-WT*) consists of *pIME1-sfGFP-IME1* followed by *C. glabrata TRP1* and the whole cassette is flanked by the 5′ and 3′ UTRs of *S. cerevisiae TRP1*. To mutate TF binding sites, DNA fragments of 500 bp in length corresponding to 701–1100 bp upstream of the *IME1* start codon with binding site mutations were synthesised (gBlocks Gene Fragments, Integrated DNA Technologies) and cloned into the *pIME1-WT* plasmid by Gibson assembly using the NEBuilder HiFi DNA Assembly Master Mix (New England BioLabs). The *pIME1-bsΔ* plasmid carried 103 mutated nucleotides (nt) between 701 and 1100 bp upstream of the *IME1* start codon to disrupt Yap6, Sok2, Phd1, Mot3, Sko1, Nrg1 and Nrg2 binding sites (Supplementary Fig. 4c). The *pIME1-spy* plasmid was designed based on the promoter sequence in *pIME1-bsΔ* by restoring the Yap6, Sok2 and Phd1 sites (33 nt) between 701–1100 bp upstream of the *IME1* start codon while other TF sites remained mutated (Supplementary Fig. 5c). Plasmid sequences were verified by Sanger sequencing. Plasmids (*pIME1-WT, pIME1-bsΔ* and *pIME1-spy*) were linearised with PmeI and integrated by transformation into the *TRP1* locus.

**Growth conditions**. Yeast cells were grown in YPD medium (1% yeast extract, 2% peptone, 2% glucose) supplemented with 96 μg/mL tryptophan, 24 μg/mL uracil and 12 μg/mL adenine, grown at 30 °C and liquid cultures were agitated at 300 r.p.m. To obtain exponentially growing cells (YPD (E)) and cells grown to saturation (YPD (S)), cells were grown in YPD to saturation overnight, diluted to OD$_{600}$ = 0.2 and subsequently YPD (E) cells were harvested after two to three doublings. YPD (S) cells were grown for 20–24 h in YPD. To induce entry into meiosis, cells were grown overnight in YPD, shifted to pre-sporulation medium BYTA (1% yeast extract, 2% tryptone, 1% potassium acetate, 50 mM potassium phthalate) at OD$_{600}$ = 0.4 for 16–18 h, and subsequently transferred to sporulation medium SPO (0.3% potassium acetate, 0.02% raffinose, pH 7.0) at OD$_{600}$ = 1.8.

To study the responses of the TFs in distinct nutrient conditions in Fig. 8b–d, cells were grown in YPD and pre-sporulation medium following the standard sporulation induction protocol. Subsequently, cells were shifted to four different types of media including sporulation medium (SPO), glucose-only medium (SPO + 2% glucose), YP medium without glucose (YP + 0.05% glucose) and YPD medium (YP + 2% glucose). Yeast cells were harvested for ChIP analyses at the point of shift (0 h SPO) and after four hours in the different nutrient conditions.

To study the effect of Tup1 and Cyc8 depletion on *IME1* expression and TF binding (Fig. 1e–i, Fig. 3b, Supplementary Figs. 2b and 4a), 500 μM of indole-3-

acetic acid (IAA) (Aldrich) was added to exponentially growing cells to induce degradation of Tup1-AID and Cyc8-AID proteins. As control, same volume of dimethyl sulphoxide (DMSO) was added to yeast cells. Cells were harvested at the indicated time points for ChIP, RT-qPCR, smFISH and western blot analyses.

**Chromatin immunoprecipitation (ChIP)**. Harvested cells were crosslinked with formaldehyde for 20 min at room temperature and reaction was quenched by the addition of 100 mM glycine. Cells were washed with FA lysis buffer (50 mM HEPES pH 7.5, 150 mM sodium chloride, 1 mM EDTA pH 7.6, 1% Triton X-100, 0.1% sodium deoxycholate, 0.1% sodium dodecyl sulphate (SDS)), snap frozen and stored at −80 °C. Cell lysis was performed in cold FA lysis buffer with cOmplete Mini Protease Inhibitor Cocktail (Roche). Samples were homogenised with zirconia beads (BioSpec) using Mini-Beadbeater-96 (BioSpec). The chromatin fraction was subjected to shearing by sonication on Bioruptor Plus (Diagenode) using 9 cycles of 30 s on, 30 s off or until the majority of fragments were under 850 bp. V5 epitope-tagged proteins were immunoprecipitated with anti-V5 agarose beads (Sigma-Aldrich) at room temperature for 2 h with rotation. Histone H3 was immuno-precipitated (Fig. 4b, Supplementary Fig. 5a) with Dynabeads Protein A (Invitrogen) coupled with anti-H3 antibodies (ab1791, Abcam) using 2 μg of antibodies per sample. Subsequently, the beads were washed with FA lysis buffer, FA lysis buffer with 260 mM sodium chloride and a lithium chloride/detergent buffer (10 mM Tris pH 8, 250 mM lithium chloride, 0.5% NP-40, 0.5% sodium deoxycholate, 1 mM EDTA). Samples were reverse crosslinked in TE buffer with 1% SDS at 65 °C, 500 r.p.m. overnight and treated with 80 μg/mL proteinase K (Thermo Scientific) at 37 °C for 2 h. Purified DNA fragments were quantified by quantitative PCR using EXPRESS SYBR GreenER SuperMix (Thermo Fisher Scientific) or PowerUp SYBR Green Master Mix (Thermo Fisher Scientific) on Applied Biosystems 7500 Fast Real-Time PCR System (Thermo Fisher Scientific). ChIP signals were typically corrected by an input, and subsequently were normalised over the silent mating type cassette *HMR*. Primer sequences are listed in Supplementary Table 1. The *HMR* locus is known to be transcriptionally inactive but binding of Tup1 has been detected near silent mating loci. To examine whether *HMR* is an appropriate background control for Tup1 ChIP, we calculated Tup1–V5 ChIP signals normalised over input for the *IME1* promoter and *HMR*, which showed that Tup1 was not enriched at the *HMR* locus (Supplementary Fig. 8).

**RNA isolation and reverse transcription**. Total RNA was extracted from harvested cells using the hot phenol method. Briefly, TES buffer (10 mM Tris-HCl pH 7.5, 10 mM EDTA, 0.5% SDS) and acid-phenol:chloroform (Ambion) were added to samples and incubated at 65 °C. RNA was subsequently purified using the NucleoSpin RNA kit (Macherey-Nagel) according to the manufacturer's instructions. rDNase was added to remove residual genomic DNA during the purification procedure. For reverse transcription, ProtoScript II First Strand cDNA Synthesis Kit (New England BioLabs) was used and 500 ng of total RNA was provided as template in each reaction. qPCR reactions were prepared using EXPRESS SYBR GreenER SuperMix (Thermo Fisher Scientific) or PowerUp SYBR Green Master Mix (Thermo Fisher Scientific) and *IME1* level was quantified on Applied Biosystems 7500 Fast Real-Time PCR System (Thermo Fisher Scientific). Signals were normalised over *ACT1*. Primer sequences are listed in Supplementary Table 1.

**Western blotting**. Proteins were extracted from cells fixed with 5% trichloroacetic acid by lysing cells in protein breakage buffer (50 mM Tris at pH 7.5, 1 mM EDTA, 27.5 mM DTT) with 0.5 mm glass beads (BioSpec) on the Mini-Beadbeater-96 (BioSpec). Samples were denatured in SDS loading buffer (62.5 mM Tris (pH 6.8), 2% β-mercaptoethanol, 10% glycerol, 3% SDS and 0.017% Bromophenol Blue) and separated by SDS-PAGE in Tris-glycine buffer (25 mM Tris base, 192 mM glycine, 0.1% SDS). V5-epitope tagged proteins were detected using anti-V5 primary antibodies (Invitrogen, 1:2000, mouse) and IRDye 800CW (LI-COR, 1:15000) or HRP-conjugated (GE Healthcare, 1:8000) secondary antibodies. For equal loading Hxk1 protein levels were determined using anti-hexokinase primary antibodies (Stratech Scientific, 1:2000, rabbit), and IRDye 680RD (LI-COR, 1:15000) or HRP-conjugated (GE Healthcare, 1:8000) secondary antibodies. Images were acquired on the Odyssey CLx imaging system (LI-COR) or by ECL Prime (GE Healthcare) using Amersham Imager 600 (GE Healthcare). Uncropped versions of the western blots presented in Fig. 1e, g, Supplementary Fig. 4a are provided in the Source Data file.

**Nuclei/DAPI counting**. Samples were fixed in 80% ethanol and stained with 1 μg/mL 4′,6-diamidino-2-phenylindole (DAPI) in PBS buffer. Cells with two, three or four DAPI masses were considered meiosis, while cells harbouring one DAPI mass were counted as NO meiosis. At least 200 cells were analysed for each sample.

**Microscopy**. For single molecule RNA fluorescence in situ hybridisation (smFISH), cells were fixed with formaldehyde and washed with Buffer B (1.2 M sorbitol, 0.1 M potassium phosphate dibasic, pH 7.5). To spheroplast the cells, samples were treated with 40 μg/mL Zymolyase-100T (MP Biomedicals) and 57.2 mM β-mercaptoethanol in Buffer B at 30 °C. Spheroplasted cells were washed and smFISH probes recognising *IME1* (AF594) and *ACT1* (Cy5)[16] were hybridised overnight in hybridisation buffer (10% dextran, 2 mM vanadyl-ribonucleoside

complex (New England BioLabs), 0.02% RNAse-free BSA, 1 mg/mL *E. coli* tRNA, 2×SSC and 10% formamide) at 30 °C. Cells were washed with wash buffer (2×SSC, 10% formamide), stained with 1 µg/mL DAPI in wash buffer and re-suspended in 2×SSC. Subsequently, cells were spun down and re-suspended prior to imaging in anti-fade GLOX with 1% catalase (Sigma) and 1% glucose oxidase (Sigma)[27]. Images were acquired using the Eclipse Ti–E inverted microscope system (Nikon) using the 100× oil objective with the ORCA-FLASH 4.0 camera (Hamamatsu) and NIS-elements software (Nikon). Cells were imaged at every 0.3 µm along the z-axis using the built-in z-axis drive and a total of 25 images were taken for each z-stack. Signals from all the planes were merged into a 2D image by applying maximum intensity z-projection in ImageJ 1.52a[60]. Only cells with *ACT1* signals were considered for *IME1* quantification using the StarSearch software (http://rajlab.seas. upenn.edu/StarSearch/launch.html).

For the quantification of sfGFP expressed in *pIME1-WT* and *pIME1-bsΔ* presented in Fig. 3d, cells were induced to sporulate using the standard protocol and samples were taken from the sporulation culture at indicated time points. Harvested cells were fixed with formaldehyde and re-suspended prior to imaging in a buffer containing 16.6 mM potassium phosphate monobasic, 83.4 mM potassium phosphate dibasic and 1.2 M sorbitol. Imaging was carried out on the Eclipse Ti–E inverted microscope system (Nikon) using the 100× oil objective with the ORCA-FLASH 4.0 camera (Hamamatsu) and NIS-elements software (Nikon). sfGFP signals in each cell were quantified using the ImageJ 1.52a software[60].

For determining the localisation of mNeonGreen-tagged TFs described in Fig. 6c. and Supplementary Fig. 6c, cells were induced to sporulate with the standard protocol. Subsequently, cells were imaged at 0 h in SPO and 4 h in SPO. Images were acquired using the same imaging system and set up described for quantification of sfGFP expressed in *pIME1-WT* and *pIME1-bsΔ* cells. Signals from whole cell and cell nucleus were quantified in the ImageJ 1.52a software and with use of the nuclear marker (2xmCherry-SV40NLS)[60]. Signal from the cytosol was inferred from the difference between whole cell signal and nuclear signal.

**IME1 promoter motif analysis**. TF binding sites presented in Fig. 2a, Supplementary Figs. 3 and 4b were predicted by scanning the *IME1* promoter with the curated TF motifs in the YeTFaSCo database (version 1.02)[61]. Binding sites were predicted to have at least 75% of the maximum possible score, with the exception of the Sut1 binding site which was predicted with a 70% threshold. The Sko1 binding site was identified manually by scanning the promoter sequence for the consensus motif TGACG as described in ref. [61].

**Statistical analyses**. Data statistics and statistical analyses indicated in the figure legends were computed using GraphPad Prism version 8.2.0 for Windows, GraphPad Software, San Diego, CA, USA, www.graphpad.com. Data from the smFISH and imaging experiments presented in Figs. 1h, 3d, 4a, c and 5a, d were analysed using unpaired parametric two-tailed Welch's *t* test with 95% confidence. The H3 ChIP data presented in Fig. 4b and Supplementary Fig. 5a and the *IME1* expression data presented in Fig. 7b were analysed by two-way ANOVA using the uncorrected Fisher's LSD method with 95% confidence. *p* Values are indicated in the figures, where ns stands for non-significant, * = ≤0.05, ** = ≤0.01, *** = ≤0.001, **** = ≤0.0001.

**Reporting summary**. Further information on research design is available in the Nature Research Reporting Summary linked to this article.

## Data availability
Supplementary Figs. 1–8 and Supplementary Table 1 are available in the supplementary information. A reporting summary for this article is available as a Supplementary Information file. The source data underlying Figs. 1b–I, 2b, c, 3b–d, 4a–c, 5a–e, 6a–c, 7a–c and 8b–d and Supplementary Figs. 1, 2b, 4a, 5a, b, 6a, c, 7a, b and 8 are provided as a Source Data file. All data are available from the corresponding author upon reasonable request.

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

## Acknowledgements

We are grateful to Andrew Wu and Fabien Moretto for their critical reading of the paper, and to Andreas Doncic for providing the mNeongreen tagging cassette. This work was supported by the Francis Crick Institute (FC001203), which receives its core funding from Cancer Research UK (FC001203), the UK Medical Research Council (FC001203) and the Wellcome Trust (FC001203).

## Author contributions

Conceptualisation: F.W. and J.T., Methodology: F.W. and J.T., Investigation: J.T., Resources: F.W. and J.T., Writing and Revisions: F.W. and J.T., Supervision: F.W., Funding acquisition: F.W.

## Competing Interests

The authors declare no competing interests.
