## [Peer Review File · Nature Communications]

Reviewers' comments:

Reviewer #1 (Remarks to the Author):

In this study the authors are investigating the regulation of IME1 transcription. IME1 encodes a master regulator which governs meiosis in the yeast *S. cerevisiae*. Meiosis in yeast is tightly regulated whereby cells will only enter into this process when specific cell and nutrient conditions are met. Indeed, the IME1 promoter is large and harbors numerous elements and transcription factor (TF) binding sites which govern its correct transcription. Prominent among these regulatory factors is the Tup1-Cyc8 complex which acts to repress IME1 transcription, and numerous other genes in yeast. Despite the Tup1-Cyc8 complex being arguably the first global repressor of transcription identified, it has not been as well studied as other regulators of transcription. Indeed, it might have been neglected due to the apparent greater interest within the scientific community of investigating how genes are activated, as opposed to how genes are repressed. In this study the authors challenge this thinking and argue that it is relief of repression from the Tup1-Cyc8 complex in response to nutrient conditions which is critical for the correct activation of the IME1 gene transcription.

The study is timely and should be interesting to a wide audience as it addresses how nutrients signal to relieve gene repression to drive transcription of a gene of the utmost importance to cell fate.

The manuscript is largely well written and the data is of good quality. The authors do shed new light on regulation of IME1 by Tup1-Cyc8.

The main findings are that:

1. Tup1 and Cyc8 bind at the same region at the repressed IME1 promoter and, following their conditional depletion, there is a rapid induction of IME1 transcription (Fig. 1).
2. The authors do a good job of identifying numerous (8) transcription factors which bind at the IME1 promoter in the vicinity of the Tup1-Cyc8 binding site and which are important for Tup1-Cyc8 occupancy (Figs. 2 and 3). The authors then present intriguing data to show that the presence of the binding sites for only three factors (Sok2, Phd1 and Yap6) is enough to maintain Tup1 at wt levels, whilst their own occupancy is reduced (Fig. 4). This result is intriguing because, unlike most models for Tup1-Cyc8 recruitment which depend on a single factor, their data suggests that Tup1 recruitment at IME1 depends on the function of numerous factors. Furthermore, the factors themselves might cooperate to reinforce their own occupancy on which Tup1-Cyc8 then resides. Thus, at IME1, it seems Tup1-Cyc8 is recruited onto a scaffold of proteins.
3. The authors then focus on three of these recruiting factors; Sok2, Phd1 and Yap6, which all contribute to repress IME1 transcription under nutrient rich conditions (Fig. 4). The fact that the de-repression of IME1 in the absence of these three TFs is not at the level found in the *tup1* mutant is entirely consistent with the other factors also being required for Tup1 occupancy and complete repression of IME1.
4. In the most important experiments (Fig. 5), the authors show that upon entry into meiosis there is a loss of multiple recruiting factors concomitant with the loss of Tup1, which the authors argue de-represses IME1 transcription (Fig 5). The authors then show that there are different dependencies of Tup1 binding upon the recruiting TFs in different nutrient conditions such that Sok2, Phd1 and Yap6 are required for Tup1 occupancy and IME1 repression under conditions that mimic the onset of meiosis (SPO media), but not under nutrient rich conditions which would not be compatible with meiosis. This is the key finding of the paper and is reinforced by data in Fig. 7, in which they present evidence to suggest that different nutrient signals deplete different recruiting factors to influence Tup1 occupancy at the IME1 promoter to yield IME1 de-repression only under the correct conditions to ensure the correct entry into meiosis.

Questions, comments and concerns:

1. Why no error bars on Fig. 1C?
2. One mechanism by which Tup1-Cyc8 has been proposed to repress genes is via the strong and orderly positioning of nucleosomes at gene promoters. If this is the case for Tup1-Cyc8 activity at IME1, the promoter truncations in Fig. 1B might be impacting IME1 transcription due to effects on nucleosome positioning as opposed to the deletion of important TF binding sites. Indeed, the only investigation in this paper into the impact of chromatin upon the mechanism of Tup1-Cyc8 regulation of IME1 transcription was to show that HDACs are not involved in repression. Considering another mechanism of action of the Tup1-Cyc8 complex is thought to be via nucleosome positioning, the absence of any analysis into histone density or chromatin structure at the IME1 gene is a noticeable omission from this study.
3. What is the lower band visible on the Tup1-AID western blots in Fig 1E and G?
4. The Fig1 legend for part G indicates 'Ime1 expression...', when it shows Tup1 protein levels.
5. In Figs 1 and 2, although the transcription factors do occupy the same site occupied by Tup1-Cyc8, the data does not definitively show that the proteins co-localise with each other (or with Tup1-Cyc8). To unequivocally show this, sequential ChIP would have to be used.
6. No error bars in Fig. 2C.
7. In Fig. 4, the 'nutrient rich' media used should be clearly stated (presumably YPD) in the legend (and text).
8. In Fig. 5A the authors show ChIP data for the recruiting TFs and Tup1 (and Pog1) in SPO media at 0 and 4h. From this they conclude that the loss of the TFs cause the loss of Tup1 from IME1 which enables transcription. Considering this is a main finding of the manuscript, I would have liked to see a finer time course of events (ChIP of Tup1 and TFs) during entry into meiosis and a clear correlation with transcription of IME1 at those same time points. Indeed, it was very hard to navigate the data shown in Fig. 5 to find out which transcription plot of which strain correlated with which ChIP plot at a given time and in a given media. For example, the ChIP data for Tup1 and the TFs at 0 and 4 h in SPO media was presented, however, only the transcription associated with time 0 in SPO was shown.
9. In Fig. 5A, why is Pog1 occupancy at 4h in SPO media reduced (significantly?), when this is the activator of IME1 which the authors state may have been 'unmasked' by the absence of Tup1? Did the authors look at any other activators of Ime1 transcription to see if they displayed a similar behaviour?
10. In Fig. 5B, Tup1 occupancy increases in the *phd1*, *sok2 phd1* and *sok2 phd1 yap6* mutants in YPD. Does this increase repression at IME1 under these conditions? The scale for the IME1 transcription plot in Fig. 5C prevents any such insight as the values for Ime1 transcription are seemingly at 0. Indeed, is this the case? Is the IME1 gene fully off in wt and all of these mutants in these conditions?
11. The symbols used in Fig. 5D were very small and difficult to tell apart from each other.
12. In Fig. 6 the authors investigate the abundance and localization of the key recruiting TFs and Tup1 and Cyc8 and also attempt to investigate the role of PKA and TORC1 signaling upon the recruiting TF protein levels. Unfortunately, I felt that this data added little to the story and, to my mind, raised more questions than answers. For example, Cyc8 levels are reduced compared to wt in SPO media and in saturated (presumably post-diauxic) YPD. Conversely, Tup1 levels remain unchanged in all conditions. Does this mean that Tup1 and Cyc8 are functioning independently of each other in these conditions? The reduction of Phd1 and Yap6 shown in Fig. 6c is not very convincing and the Yap6 signal is poor to start with.

Reviewer #2 (Remarks to the Author):

The IME1 gene plays a critical role in regulating the decision of whether to enter the meiotic program. The gene has an exceptionally large promoter and complex regulation. This paper attempts to provide new insights as to how IME1 is regulated. While there is a great deal of data in this paper, there are relatively few findings of significance, and no mechanistic insights.

It has been known for a while that there are many transcriptional regulators that could bind to IME1, from the genetic analyses of Kahana et al (2010). Here the authors perform ChIP experiments to show that eight transcription factors do in fact bind to IME1. The authors had previously shown that the TUP1 co-repressor is recruited to IME1, and here they show that three factors, Sok2, Phd1, and Yap6, all recruit Tup1 to IME1. While this is a result, it is hardly of significance to the field as the literature is rife with examples of DNA-binding proteins recruiting Tup1 to promoters as a repressive mechanism. Finally, we learn nothing new about the mechanisms that overcome Tup1 repression at IME1.

Specific comments.

1. A major problem I have with this paper is the authors repeated over-arching conclusions that are not supported by the data. Page 8. "We conclude that there is little to no temporal delay between Tup1-Cyc8 depletion and IME1 expression ..." Page 21. "Remarkably, there was little or no delay between depletion of Tup1 and IME1 transcription in the presence of ample nutrients." These statements are way too strong based on the data presented.
2. On Page 9, after discussing the results with two HDAC double mutants, the authors "conclude that HDACs that are known to interact with Tup1- Cyc8 play only a marginal role in repressing the IME1 promoter." It is possible, even likely, that there is significant redundancy and a triple or quadruple HDAC mutant might be needed to see an effect.
3. I have critical questions about the ChIP quantitation, particularly for the data in Figure 2. In the Methods it says "ChIP signals were normalised over the silent mating type cassette HMR." I have questions: Were they also normalized to input, as a control for DNA yield? Was the HMR normalization done for all samples? For Tup1 ChIPs, normalization to HMR serves as a positive control, while for the DNA-binding proteins it is a negative control. On page 10 it says that for some factors "binding was above background." How was this determined?
4. The question of ChIP quantitation and normalization is also important for the data in Figure 5A, and how that data can be interpreted. Some binding is described as marginal, but the experiment is lacking a control for the untagged control. Similarly, the description of Figure 5B says "Strikingly, Tup1 association with the IME1 promoter was reduced to nearly background levels in sok2Δ yap6Δ and sok2Δ phd1Δ yap6Δ cells at 0 hours in SPO." Without an untagged control, we do not know what are background levels. I do note that an untagged control is presented in the experiment in Figure 2B. It is not at all clear from the legend whether the "control" in Figure 3C is an untagged control.
5. Page 18. "We conclude that inhibiting PKA and TORC1 affects the abundance of transcription factors important for repressing the IME1 promoter, and coincides with Tup1 disassociation and activation of IME1 transcription as described previously." The authors should perform a simple experiment that could support their contention, and that is a Tup1-V5 ChIP at 0 hours in Spo.
6. Page 22. The authors say "transcriptional activators do not require nutrient or environmental signalling to activate the IME1 promoter." You cannot make this conclusion, as the activator is unknown. In fact, later in the paragraph the authors state "Several transcriptional activators have been implicated in regulating IME1 transcription that have not been linked with Tup1-Cyc8."
7. Page 22. "Furthermore, we show that the transcription activator Pog1 is bound to the IME1 promoter prior to activation, and remains bound during activation of IME1 transcription." However, you do not know whether Pog1 is the sole activator required.
8. Page 13. "Furthermore, Yap6, Sok2, and Phd1 were enriched at the IME1 promoter in pIME1-spy cells but their binding was reduced compared to the wild-type promoter - suggesting that

there are additional binding sites (Figure 4C).” It seems equally plausible that the factor bindings are truly eliminated, but cooperative interactions with other factors still binding to IME1 can recruit Yap6, Sok2, and Phd1.

9. Page 15. “We found that Tup1 binding to the IME1 promoter was not affected in rich medium containing glucose in sok2, phd1, and yap6 single/double/triple deletion mutants ...” I disagree. I see increased Tup1 binding in phd1 mutants.

10. Statistical Analysis. In the authors’ reporting summary, they state that “The exact sample size (n) for each experimental group/condition, given as a discrete number and unit of measurement” in response to the question whether “For all statistical analyses, confirm that the following items are present in the figure legend, table legend, main text, or Methods section.” This is clearly not true. For example, Figure 1D and 1F show dots in the figure, which might mean duplicate or triplicate, but NO information as to number of replicates is provided in the figure legend.

11. Similarly, how many replicates in Figure 3B? Is the increase in Nrg1-V5 statistically significant? If so, what does it mean biologically?

12. In Figure 3 the authors use a pIME1-bs Δ construct with mutations in factor binding sites. ChIPs should be performed to address whether the sequence changes truly eliminate factor binding.

13. In discussing Figure 1H and 1I, the authors state “It is worth noting that the AID-tag fused to Tup1 had some effect on IME1 expression in the absence of IAA as IME1 transcript levels were increased by five-fold in TUP1-AID compared to wild-type cells.” I see no data showing a comparison of the Tup1-AID tagged strain to wild type.

14. Page 19. “In agreement with this model, IME1 expression was only marginally increased in sok2 Δ phd1 Δ yap6 Δ cells grown in the presence of ample nutrients with glucose as the carbon source (YPD), while the IME1 promoter was nearly completely de-repressed in sok2 Δ phd1 Δ yap6 Δ cells grown in acetate containing medium (Figure 4A, 4B, 5B and 5C).” Figures 4A and 4B do not show complete depression, and Figure 5B is a ChIP experiment.

15. Page 8. “It is worth noting that the AID-tag fused to Tup1 had some effect on IME1 expression in the absence of IAA as IME1 transcript levels were increased by five-fold in TUP1-AID compared to wild-type cells (Figure 1H, 1I, and Supplementary Figure 2B and 2C).” I do not see why Supplementary Figure 2C is mentioned here.

16. The western blot in Supplementary Figure 4 has a band marked by an asterisk. I assume this is a cross reacting species, but it should be noted in the legend.

17. Page 13. “We found that Tup1 binding to the IME1 promoter was not affected in rich medium containing glucose in sok2, phd1, and yap6 single/double/triple deletion mutants, which is in line with IME1 expression data described in (Figure 5B, 4A, and 4B).” I think the figure references are misplaced, and the sentence should read: “We found that Tup1 binding to the IME1 promoter was not affected in rich medium containing glucose in sok2, phd1, and yap6 single/double/triple deletion mutants (Figure 5B), which is in line with IME1 expression data described in Figures 4A and 4B.

18. Page 18. “Therefore, perhaps inhibiting PKA and TORC1 altogether could reveal how both signalling pathways regulate transcription factors important for repressing the IME1 promoter.” I think “altogether” should be “together.”

Reviewer #3 (Remarks to the Author):

Tan and van Werven set out to determine how the expression of Ime1--the master regulator of meiosis in yeast--is regulated. Using a series of well-designed and well-executed experiments the authors show that Ime1 expression is controlled by at least eight transcription factors that serve to recruit the repressor proteins Tup1 and Cyc8 to the -1200 to -800 region of the Ime1 promoter. These TFs are then released from the promoter in starvation conditions, leading to release of Tup1/Cyc8 and de-repression of Ime1. Three of these transcription factors (Sok2, Phd1 and Yap6) drive gene repression in glucose starvation conditions, while other factors such as Nrg2 and Mot3 act redundantly with Sok2, Phd1 and Yap6 in glucose replete conditions. Together, the data presented, convincingly show that a large group of TFs act together at the Ime1 promoter, and are regulated by distinct starvation signals to ensure that Ime1 is only de-repressed, and meiosis initiated, when cells experience complete starvation.

The authors also carry out experiments trying to link the clearance of Sok2, Phd1 and Yap6 from the Ime1 promoter to the starvation signals that initiate meiosis. They find that the TFs are stable in conditions that induce meiosis and remain in the nucleus. However, when they use chemical inhibitors to block PKA or TORC1 signaling (events known to trigger meiosis) they do see degradation of Sok2, Phd1 and Yap6. Based on these data they state that PKA and TORC1 may act locally at the promoter to control the degradation of the Sok2, Phd1 and Yap6 factors. This argument does not make any sense to me. The PKA kinases have been shown to bind to promoter regions (in contrast all recent data suggest that TORC1 is localized on the vacuole and late endosomes), but is the argument that only the TFs on the promoter are dephosphorylated and degraded? It seems far more likely that four hours of complete Tpk1 inhibition does lead to Sok2, Phd1 and Yap6 degradation, perhaps to ensure Ime1 derepression in extreme starvation, but that is not the mechanism at play when there is low level PKA signaling as expected in the SPO medium.

Overall then Tan et al offers a high-quality characterization of TF action at the Ime1 promoter, and shows that groups of TFs act redundantly to recruit Tup1/Cyc8 and keep the gene off until the conditions are right for meiosis. This is an interesting finding, but it is not clear to me that it clears the bar for publication in Nature Communications. As the authors themselves point out, there are already examples of multiple TFs acting redundantly to control transcription through repressor complexes, so this is ultimately an Ime1 and yeast specific story.

Comments

(1) It is worth mentioning that the quality of the data and analysis are excellent, especially the FISH data used throughout to measure Ime1 repression in single cells.

(2) As mentioned above the argument that PKA and TORC1 "locally regulate" the Ime1 promoter is confusing. The authors should explain how this could work, or alter this part of the paper.

(3) Minor point: The authors state that five of the TFs stop binding after 4 hrs in SPO medium, but it only looks like three stop binding to me. What is the cutoff and do they consider the fold difference between 0 hrs SPO and 4hrs SPO for all the factors, they are probably identical (that is there is fractional weak binding of all factors and some drop below the measurable threshold).

(4) Minor point: why are the authors using Hxk1 as a loading control? Don't Hxk1 levels change in glucose starvation.

Reviewers' comments:

Reviewer #1 (Remarks to the Author):

Questions, comments and concerns:

1. Why no error bars on Fig. 1C?

We have added datapoints and now show two biological repeats.

2. One mechanism by which Tup1-Cyc8 has been proposed to repress genes is via the strong and orderly positioning of nucleosomes at gene promoters. If this is the case for Tup1-Cyc8 activity at *IME1*, the promoter truncations in Fig. 1B might be impacting *IME1* transcription due to effects on nucleosome positioning as opposed to the deletion of important TF binding sites. Indeed, the only investigation in this paper into the impact of chromatin upon the mechanism of Tup1-Cyc8 regulation of *IME1* transcription was to show that HDACs are not involved in repression. Considering another mechanism of action of the Tup1-Cyc8 complex is thought to be via nucleosome positioning, the absence of any analysis into histone density or chromatin structure at the *IME1* gene is a noticeable omission from this study.

We thank the reviewer for this comment. Indeed, the Tup1-Cyc8 co-repressor complex has been described to affect nucleosome positioning. In fact, previous work showed that a large part of the *IME1* promoter is devoid of nucleosomes in the absence of Tup1, suggesting that Tup1-Cyc8 plays a role in regulating nucleosome occupancy (Figure 8a in Weidberg et al)^{1, 2}. However, it is difficult to discriminate between direct and indirect effects on chromatin structure when Tup1-Cyc8 is depleted. As soon as Tup1-Cyc8 is depleted, transcriptional activators can come in to activate gene expression, which in turn change the chromatin structure. There is already good evidence that Tup1-Cyc8 can repress gene promoters by blocking transcriptional activators and largely independent of regulating the chromatin state. For example, by time-course experiments, it has been demonstrated that Tup1-Cyc8 mediated repression occurs prior to establishing nucleosome changes suggesting that nucleosome changes is a secondary effect³.

We have now examined the chromatin structure in cells harbouring the wild-type *IME1* promoter, and the TF-*bsΔ* mutant. This mutant displays a strong reduction in Tup1 binding because the sequence specific transcription factor binding motifs were mutated. In addition to effect on Tup1 recruitment, the TF-*bsΔ* mutant is also less active and thus affects *IME1* promoter activity. We found that there were little differences in histone H3 occupancy between WT and TF-*bsΔ* mutant in YPD suggesting that Tup1 does not play a direct role in preventing nucleosome depletion or eviction (Figure 4b). It is possible that there are effects on nucleosome positioning that we cannot detect by the ChIP assay. Nevertheless, our analysis suggests that Tup1-Cyc8 does not directly prevent nucleosome eviction of the *IME1* promoter. We have re-organised the manuscript and have now devoted Figure 4 to present the chromatin related data of the manuscript.

3. What is the lower band visible on the Tup1-AID western blots in Fig 1E and G?

The extra band is not a background band caused by antibody cross reactivity. The extra band is also not sensitive to auxin and so it is likely to have been cleaved from the C-terminus where the AID degron is. This truncated Tup1 protein is not part of a functional Tup1-Cyc8 complex because we cannot ChIP it at the *IME1* promoter when the main form of Tup1-AID is depleted (Figure 3b). In addition, the *IME1* promoter is fully de-repressed in the presence of auxin when the truncated Tup1 protein is expressed (Figure 1f). Thus, the protein product

represented by the band is not functional. We have labelled the band in the figure legend of Figure 1e.

4. The Fig1 legend for part G indicates ‘Ime1 expression...’, when it shows Tup1 protein levels.

Thank you, we have corrected the labelling error.

5. In Figs 1 and 2, although the transcription factors do occupy the same site occupied by Tup1-Cyc8, the data does not definitively show that the proteins co-localise with each other (or with Tup1-Cyc8). To unequivocally show this, sequential ChIP would have to be used.

We thank the reviewer for this comment. While we agree that sequential ChIP would be a great experiment to add to the manuscript (if it works), this technique is also notoriously difficult and only a few labs have successfully used this. In the past, we have attempted to setup the technique in my laboratory but we were not successful. Hence, we cannot provide data on sequential ChIP.

We agree that Figure 1 and 2 give correlative evidence. Previous work by others has shown that some of these transcription factors can physically interact with Tup1-Cyc8 in vitro. The data in figures 3, 5, 7, however, demonstrate the interactions between the *IME1* promoter sequence, transcription factors and Tup1-Cyc8. These data strongly suggest that Tup1-Cyc8 and these transcription factors co-localize at the *IME1* promoter, and recruit Tup1-Cyc8 to the *IME1* promoter.

Please note, we changed the statement that the transcription factor co-localized on page 11 to the following:

The binding of these transcription factors peaked in the same region of the *IME1* promoter as Tup1-Cyc8 at around 1000 bp upstream of the *IME1* start codon.

6. No error bars in Fig. 2C.

We have added the datapoints showing two biological repeats. Since, showing all the data points made the combined graph unreadable, we formatted the data for each transcription factor in separate graphs.

7. In Fig. 4, the ‘nutrient rich’ media used should be clearly stated (presumably YPD) in the legend (and text).

We have now explicitly stated that we grew cells in YPD till exponential growth phase.

8. In Fig. 5A the authors show ChIP data for the recruiting TFs and Tup1 (and Pog1) in SPO media at 0 and 4h. From this they conclude that the loss of the TFs cause the loss of Tup1 from *IME1* which enables transcription. Considering this is a main finding of the manuscript, I would have liked to see a finer time course of events (ChIP of Tup1 and TFs) during entry into meiosis and a clear correlation with transcription of *IME1* at those same time points. Indeed, it was very hard to navigate the data shown in Fig. 5 to find out which transcription plot of which strain correlated with which ChIP plot at a given time and in a given media. For example, the ChIP data for Tup1 and the TFs at 0 and 4 h in SPO media was presented, however, only the transcription associated with time 0 in SPO was shown.

To gain further insight into how transcription factors and Tup1 dissociate from the *IME1* promoter, we performed a time course experiment with three transcription factors (Yap6, Sok2, and Phd1) and Tup1 (Figure 6b). We found that *IME1* bulk expression levels peaked at

3 hours in SPO (Supplementary Fig. 6a). At the same time, Tup1 and Sok2 dissociation from the *IME1* promoter was detected in the early time points (1 and 2 hours in SPO), while the majority of Tup1 and Sok2 was evicted at 4 hours in SPO (Fig. 6b). Yap6 and Phd1 displayed reduced binding at the *IME1* promoter at 3 hours in SPO and was thus somewhat slower than Sok2 and Tup1.

9. In Fig. 5A, why is Pog1 occupancy at 4h in SPO media reduced (significantly?), when this is the activator of *IME1* which the authors state may have been 'unmasked' by the absence of Tup1? Did the authors look at any other activators of *Ime1* transcription to see if they displayed a similar behaviour?

Pog1 binding is lower (but still clearly enriched) during activation of the *IME1* promoter. We can only speculate about the reason for that. One possibility is that in the absence of Tup1-Cyc8, the ability for Pog1 to crosslink with chromatin is reduced. Another possibility is that Pog1 is more dynamic under activating conditions. It will require in depth investigations to determine how the ability of Pog1 to bind the *IME1* promoter is affected under active and repressive conditions, which is beyond the scope of the current manuscript.

Apart from Pog1, no other transcriptional activator has been characterised, but many have been implicated. We are currently examining multiple other candidate transcriptional activators to find further support for our model. We have preliminary data suggesting that other transcription activators also bind to the *IME1* promoter under repressing conditions. However, more work is needed to characterise how they function at the *IME1* promoter, which we believe is beyond the scope of the current manuscript.

10. In Fig. 5B, Tup1 occupancy increases in the *phd1*, *sok2 phd1* and *sok2 phd1 yap6* mutants in YPD. Does this increase repression at *IME1* under these conditions? The scale for the *IME1* transcription plot in Fig. 5C prevents any such insight as the values for *Ime1* transcription are seemingly at 0. Indeed, is this the case? Is the *IME1* gene fully off in wt and all of these mutants in these conditions?

In figure 5b (now figure 7a), some of the mutants indeed show an increase in Tup1 binding in YPD compared to the WT. We have analysed the *IME1* expression of these mutants in figure 5a and 5b by smFISH and found that there was an increase of *IME1* expression in the *sok2Δphd1Δ* and *sok2Δphd1Δyap6Δ* mutants in a small fraction of cells. Thus, the increased binding does not reflect more repression under YPD conditions, rather a marginal increase in *IME1* expression. This suggests that more binding of Tup1 perhaps does not reflect the ability of Tup1 to repress, while less binding does correlate with de-repression. One possibility is that Tup1 crosslinks better at the *IME1* promoter in the mutants because the absence of several transcription factors causes the proximity between Tup1-Cyc8 and *IME1* promoter to decrease.

11. The symbols used in Fig. 5D were very small and difficult to tell apart from each other. We have changed the symbols, we think the figure is clearer now.

12. In Fig. 6 the authors investigate the abundance and localization of the key recruiting TFs and Tup1 and Cyc8 and also attempt to investigate the role of PKA and TORC1 signaling upon the recruiting TF protein levels. Unfortunately, I felt that this data added little to the story and, to my mind, raised more questions than answers. For example, Cyc8 levels are reduced compared to wt in SPO media and in saturated (presumably post-diauxic) YPD. Conversely, Tup1 levels remain unchanged in all conditions. Does this mean that Tup1 and

Cyc8 are functioning independently of each other in these conditions? The reduction of Phd1 and Yap6 shown in Fig. 6c is not very convincing and the Yap6 signal is poor to start with. We have removed the western blot data and PKA and TORC1 data from the manuscript. We decided to keep the localization data in the manuscript because it excludes an obvious explanation of how these transcription factors dissociate. In addition, we reorganized the data of Figure 5 and 6 (now Figure 6 and 7). Overall, we think this has improved the clarity of the manuscript.

Reviewer #2 (Remarks to the Author):

Specific comments.

1. A major problem I have with this paper is the authors repeated over-arching conclusions that are not supported by the data. Page 8. “We conclude that there is little to no temporal delay between Tup1-Cyc8 depletion and IME1 expression ...” Page 21. “Remarkably, there was little or no delay between depletion of Tup1 and IME1 transcription in the presence of ample nutrients.” These statements are way too strong based on the data presented.

We agree that these statements were perhaps a bit strong. However, we think that the observations in figure 1h and 1i clearly show that as soon as Tup1 is depleted IME1 transcripts start to accumulate in single cells. To really know whether there is a delay or not, measuring transcription (as supposed to RNA accumulation) may be more appropriate at the single cell level, which is quite difficult.

We have rephrased the statements into the following:

Our analysis indicates that there is little temporal delay between Tup1-Cyc8 depletion and IME1 transcript accumulation suggesting that transcriptional activators are bound or readily available for recruitment to the IME1 promoter.

Remarkably, we detected little delay between depletion of Tup1 and de-repression of IME1 expression in the presence of ample nutrients (Fig. 1g, 1h and 1i).

2. On Page 9, after discussing the results with two HDAC double mutants, the authors “conclude that HDACs that are known to interact with Tup1- Cyc8 play only a marginal role in repressing the IME1 promoter.” It is possible, even likely, that there is significant redundancy and a triple or quadruple HDAC mutant might be needed to see an effect.

We agree with that it is indeed possible. However, repression of certain loci, like the FLO1 promoter, requires two HDACs (Hda1 and Rpd3)⁴. This was clearly not the case for IME1. We have not further examined triple and quadruple mutant because these mutants tend to have slow growth phenotype, which could cause other secondary effects. Therefore, we have adjusted our statement into the following:

Our data suggest HDACs that are known to interact with Tup1-Cyc8 play only a marginal role in repressing the IME1 promoter. It is worth noting that our analysis does not exclude the possibility there is further redundancy between HDACs in regulating the IME1 promoter.

3. I have critical questions about the CHIP quantitation, particularly for the data in Figure 2. In the Methods it says “CHIP signals were normalised over the silent mating type cassette

HMR.” I have questions: Were they also normalized to input, as a control for DNA yield? Was the HMR normalization done for all samples? For Tup1 ChIPs, normalization to HMR serves as a positive control, while for the DNA-binding proteins it is a negative controls. On page 10 it says that for some factors “binding was above background.” How was this determined?

Yes, the normalization over HMR was done for all samples including Tup1. We used an input sample for determining the overall signal, and subsequently use HMR to normalize over background. Furthermore, we include untagged control strain for most experiments throughout the manuscript. While there is evidence that Tup1 associates with the silent mating type loci, the region for our control primers did not show enrichment for Tup1 when normalized over input (supplementary Figure 8). Therefore, this region of the HMR locus is an appropriate background control for the Tup1 ChIP.

The control for our analysis in figure 2 was an untagged strain. We have defined the binding above background in a clearer way and stated this in the text or legends where appropriate. Typically, we defined 3-fold or more over HMR as the cut-off.

We also have revised the statement into the following:

The transcription factors Mot3, Sko1, Nrg1, and Nrg2 displayed a milder enrichment, but their binding was above background levels (3-fold or more over background).

4. The question of ChIP quantitation and normalization is also important for the data in Figure 5A, and how that data can be interpreted. Some binding is described as marginal, but the experiment is lacking a control for the untagged control. Similarly, the description of Figure 5B says “Strikingly, Tup1 association with the *IME1* promoter was reduced to nearly background levels in *sok2Δ yap6Δ* and *sok2Δ phd1Δ yap6Δ* cells at 0 hours in SPO.” Without an untagged control, we do not know what are background levels. I do note that an untagged control is presented in the experiment in Figures 2B. It is not at all clear from the legend whether the “control” in Figure 3C is an untagged control.

We have now included an untagged control in figure 6a (previously figure 5a) and an untagged control to figure 7a (previously figure 5b). This demonstrates clearly that some of the mutants in figure 7a show close to background signals. The control for figure 2c (not Figure 3c) is indeed an untagged control. We have now made this clear in the figure. We have changed the statement into:

“Strikingly, Tup1 association with the *IME1* promoter was severely reduced (less than 3-fold over background) in *sok2Δyap6Δ* and *sok2Δphd1Δyap6Δ* cells at 0 hours in SPO.”

5. Page 18. “We conclude that inhibiting PKA and TORC1 affects the abundance of transcription factors important for repressing the *IME1* promoter, and coincides with Tup1 disassociation and activation of *IME1* transcription as described previously.” The authors should perform a simple experiment that could support their contention, and that is a Tup1-V5 ChIP at 0 hours in *Spo*.

We did not understand the suggested experiment. Since reviewer 1 suggested to take out the PKA and TORC1 data, we have removed the PKA and TORC1 panels from the manuscript. Hence, the analysis in relation to PKA and TORC1 signalling is not relevant to the revised version of the manuscript.

6. Page 22. The authors say “transcriptional activators do not require nutrient or environmental signalling to activate the *IME1* promoter.” You cannot make this conclusion, as the activator is unknown. In fact, later in the paragraph the authors state “Several transcriptional activators have been implicated in regulating *IME1* transcription that have not been linked with Tup1-Cyc8.”

Given that Tup1 or Cyc8 depletion de-represses the *IME1* promoter to high levels in rich nutrients conditions (YPD), it is clear that key activators are active in nutrient rich conditions. Indeed, formally we cannot exclude the possibility that some transcription factors are regulated by nutrient signalling. Therefore, we have weakened the statement to the following:

“Second, key transcriptional activators of the *IME1* promoter can be active under nutrient rich conditions when Tup1-Cyc8 is unbound.”

7. Page 22. “Furthermore, we show that the transcription activator Pog1 is bound to the *IME1* promoter prior to activation, and remains bound during activation of *IME1* transcription.” However, you do not know whether Pog1 is the sole activator required.

Pog1 is not the sole activator and indeed it remains to be determined how other activators associate. We reformulated this statement to the following:

“Furthermore, we show that the transcription activator Pog1 is bound to the *IME1* promoter prior to activation and remains bound during activation of *IME1* transcription (Fig. 1d and 6a). However, Pog1 is not the only transcriptional activator of the *IME1* promoter. Thus, more work is needed to identify and dissect how other transcriptional activators regulate the *IME1* promoter.”

8. Page 13. “Furthermore, Yap6, Sok2, and Phd1 were enriched at the *IME1* promoter in *pIME1-spy* cells but their binding was reduced compared to the wild-type promoter - suggesting that there are additional binding sites (Figure 4C).” It seems equally plausible that the factor bindings are truly eliminated, but cooperative interactions with other factors still binding to *IME1* can recruit Yap6, Sok2, and Phd1.

Thank you for the suggestion, we have altered the statement to the following:

“Furthermore, Yap6, Sok2, and Phd1 were enriched at the *IME1* promoter in *pIME1-spy* cells but their binding was reduced compared to the wild-type promoter - suggesting that additional binding sites are present or cooperative interactions with remaining transcription factors exist (Fig. 5c).”

9. Page 15. “We found that Tup1 binding to the *IME1* promoter was not affected in rich medium containing glucose in *sok2*, *phd1*, and *yap6* single/double/triple deletion mutants ...” I disagree. I see increased Tup1 binding in *phd1* mutants.

We have changed the statement into the following:

“We found that Tup1 binding to the *IME1* promoter was not decreased in *sok2*, *phd1*, and *yap6* single/double/triple deletions in rich medium containing glucose (Fig. 7a).”

10. Statistical Analysis. In the authors’ reporting summary, they state that “The exact sample size (n) for each experimental group/condition, given as a discrete number and unit of measurement” in response to the question whether “For all statistical analyses, confirm that

the following items are present in the figure legend, table legend, main text, or Methods section.” This is clearly not true. For example, Figure 1D and 1F show dots in the figure, which might mean duplicate or triplicate, but NO information as to number of replicates is provided in the figure legend.

We have now stated the number of replicates in the figure legends for each sample and experiment.

11. Similarly, how many replicates in Figure 3B? Is the increase in Nrg1-V5 statistically significant? If so, what does it mean biologically?

We have repeated the experiment, and now included the data of two biological replicates. There was a very marginal increase in Nrg1 binding, which is unlikely biologically relevant.

12. In Figure 3 the authors use a pIME1-bs Δ construct with mutations in factor binding sites. ChIPs should be performed to address whether the sequence changes truly eliminate factor binding.

We agree, we have performed such an experiment in figure 5c. We show that Tup1, Sok2, and Yap6 show near background binding in the pIME1-bs mutant (less than 3-fold over background), while Phd1 was slightly above background (less than 4-fold over background).

13. In discussing Figure 1H and 1I, the authors state “It is worth noting that the AID-tag fused to Tup1 had some effect on IME1 expression in the absence of IAA as IME1 transcript levels were increased by five-fold in TUP1-AID compared to wild-type cells.” I see no data showing a comparison of the Tup1-AID tagged strain to wild type.

We agree, that this control was not clearly presented. We have now included a WT control in figure 1H.

14. Page 19. “In agreement with this model, IME1 expression was only marginally increased in sok2 Δ phd1 Δ yap6 Δ cells grown in the presence of ample nutrients with glucose as the carbon source (YPD), while the IME1 promoter was nearly completely de-repressed in sok2 Δ phd1 Δ yap6 Δ cells grown in acetate containing medium (Figure 4A, 4B, 5B and 5C).”

Figures 4A and 4B do not show complete depression, and Figure 5B is a ChIP experiment.

We have rephrased the sentence:

“In agreement with this model, *IME1* expression was only marginally increased in sok2 Δ phd1 Δ yap6 Δ cells grown in the presence of ample nutrients with glucose as the carbon source (YPD), while *IME1* expression was increased by nearly 10-fold in sok2 Δ phd1 Δ yap6 Δ cells grown in acetate-containing medium (Figure 5a, 5b, and 7b).”

15. Page 8. “It is worth noting that the AID-tag fused to Tup1 had some effect on IME1 expression in the absence of IAA as IME1 transcript levels were increased by five-fold in TUP1-AID compared to wild-type cells (Figure 1H, 1I, and Supplementary Figure 2B and 2C).” I do not see why Supplementary Figure 2C is mentioned here.

For clarity, we have added the wild-type control in figure 1h.

16. The western blot in Supplementary Figure 4 has a band marked by an asterisk. I assume this is a cross reacting species, but it should be noted in the legend.

The extra band is not a background band caused by antibody cross reactivity. The extra band

is also not sensitive to auxin and so it is likely to have been cleaved from the C-terminus where the AID degron is. This truncated Tup1 protein is not part of a functional Tup1-Cyc8 complex because we cannot ChIP it at the *IME1* promoter when the main form of Tup1-AID is depleted. In addition, the *IME1* promoter is fully de-repressed in the presence of auxin when the truncated Tup1 protein is expressed. Thus, the protein product represented by the band is not functional, and therefore does not affect the interpretation of the experiments. We have labelled the band in the figure legend as Tup1-AID cleavage product.

17. Page 13. “We found that Tup1 binding to the *IME1* promoter was not affected in rich medium containing glucose in *sok2*, *phd1*, and *yap6* single/double/triple deletion mutants, which is in line with *IME1* expression data described in (Figure 5B, 4A, and 4B).” I think the figure references are misplaced, and the sentence should read: “We found that Tup1 binding to the *IME1* promoter was not affected in rich medium containing glucose in *sok2*, *phd1*, and *yap6* single/double/triple deletion mutants (Figure 5B), which is in line with *IME1* expression data described in Figures 4A and 4B.

We fixed the reference accordingly.

18. Page 18. “Therefore, perhaps inhibiting PKA and TORC1 altogether could reveal how both signalling pathways regulate transcription factors important for repressing the *IME1* promoter.” I think “altogether” should be “together.”

We have removed this part from the manuscript. Therefore, the suggestion is not relevant to the revised manuscript.

Reviewer #3 (Remarks to the Author):

Comments

(1) It is worth mentioning that the quality of the data and analysis are excellent, especially the FISH data used throughout to measure *Ime1* repression in single cells.

Thank you.

(2) As mentioned above the argument that PKA and TORC1 “locally regulate” the *Ime1* promoter is confusing. The authors should explain how this could work, or alter this part of the paper.

We have altered the PKA and TORC part of the paper as also suggested by reviewer 1. We have removed the western blot data.

(3) Minor point: The authors state that five of the TFs stop binding after 4 hrs in SPO medium, but it only looks like three stop binding to me. What is the cutoff and do they consider the fold difference between 0 hrs SPO and 4hrs SPO for all the factors, they are probably identical (that is there is fractional weak binding of all factors and some drop below the measurable threshold).

Indeed, three transcription factors (*Sko1*, *Nrg2*, and *Sut1*) display very similar signals as the untagged control. *Sok2* and *Mot3* signals were slightly higher but have less than three-fold binding over background which was what we used as a cut-off. We have made this clear in the text by incorporating the following statement:

“Strikingly, five transcription factors showed near background binding (less than 3-fold) to the *IME1* promoter upon entry into meiosis (4 hours in SPO),”

(4) Minor point: why are the authors using Hxk1 as a loading control? Don't Hxk1 levels change in glucose starvation.

We have removed these data from the current manuscript as suggested by reviewer 1, hence the comment is no longer relevant for the revised manuscript.

References:

1. Weidberg, H., Moretto, F., Spedale, G., Amon, A. & van Werven, F.J. Nutrient Control of Yeast Gametogenesis Is Mediated by TORC1, PKA and Energy Availability. *PLoS Genet* **12**, e1006075 (2016).
2. Rizzo, J.M., Mieczkowski, P.A. & Buck, M.J. Tup1 stabilizes promoter nucleosome positioning and occupancy at transcriptionally plastic genes. *Nucleic Acids Res* **39**, 8803-8819 (2011).
3. Wong, K.H. & Struhl, K. The Cyc8-Tup1 complex inhibits transcription primarily by masking the activation domain of the recruiting protein. *Genes Dev* **25**, 2525-2539 (2011).
4. Fleming, A.B., Beggs, S., Church, M., Tsukihashi, Y. & Pennings, S. The yeast Cyc8-Tup1 complex cooperates with Hda1p and Rpd3p histone deacetylases to robustly repress transcription of the subtelomeric FLO1 gene. *Biochim Biophys Acta* **1839**, 1242-1255 (2014).

REVIEWERS' COMMENTS:

Reviewer #1 (Remarks to the Author):

This is the revised version of a manuscript that I had previously reviewed. In my original review, I had listed concerns and queries that the authors have now addressed to my satisfaction.

Reviewer #2 (Remarks to the Author):

The authors have addressed all of my concerns.

I did find one word needing correction: On page 5 line 123 the authors state "In short, we found that regulated repression by multiple sequence specific transcription factors mediating the association of Tup1-Cyc8 with the IME1 promoter is the mean by which IME1 transcription is controlled." Rather than "the mean" it should be "the means."

Reviewer #3 (Remarks to the Author):

The authors have answered all of my questions, and have done an excellent job of revising the manuscript in general. It is ready for publication.

Response to REVIEWERS' COMMENTS:

Reviewer #1 (Remarks to the Author):

This is the revised version of a manuscript that I had previously reviewed. In my original review, I had listed concerns and queries that the authors have now addressed to my satisfaction.

Reviewer #2 (Remarks to the Author):

The authors have addressed all of my concerns.

I did find one word needing correction: On page 5 line 123 the authors state “In short, we found that regulated repression by multiple sequence specific transcription factors mediating the association of Tup1-Cyc8 with the IME1 promoter is the mean by which IME1 transcription is controlled.” Rather than “the mean” it should be “the means.”

We have corrected this accordingly.

Reviewer #3 (Remarks to the Author):

The authors have answered all of my questions, and have done an excellent job of revising the manuscript in general. It is ready for publication.